# Sentiment analysis of classical Chinese literature: An unsupervised deep learning model with BERT and graph attention networks

Xiaohan Yu[1], Jin Wang [ID][2]*

1 School of Arts and Social Sciences, Hong Kong Metropolitan University, Kowloon, Hong Kong, 2 School of Foreign Languages, Weifang University, Weifang, China

* jkxr58@163.com

## Abstract

Sentiment analysis has become a transformative technology in various contexts, particularly in Natural Language Processing (NLP), social media analytics, and literary analysis, as it can extract information from a wide range of texts. The advancements in deep learning, particularly with transformer models such as BERT and graph-based models like GATs, have enabled faster progress in analyzing complex language structures. However, the issue lies in incorporating these technologies into classical Chinese literature, which involves delicate syntax, semantics, and emotions that are difficult to harness using traditional methods. The existing methods, which rely on strictly labeled data or unsupervised learning methods that do not effectively manage contextual dependencies, are very limited in analyzing historical or philosophical texts that abound in metaphor and implicit sentiment. To minimize the limitations, this paper proposes an unsupervised deep learning framework that integrates BERT embeddings, sentiment lexicon enrichment, and graph attention networks (GATs) for sentiment analysis in classical Chinese literature. Firstly, the BERT-based model extracts contextualised embeddings from a raw text, providing a deep understanding of semantics. Secondly, embedding includes sentiment-specific data from the NTUSD lexicon, thus injecting it with emotional information. Thirdly, a graph-based formulation is developed, in which words are represented as nodes, and the relations between them are defined using GATs to modify the features of nodes based on their significance in the context. Finally, unsupervised sentiment labelling, or K-Means clustering, is used to classify sentiment. The experimental results demonstrate the proposed model's efficiency – an accuracy of 0.95, precision of 0.97, recall of 0.96, and F1-score of 0.91 in several runs. These results surpass those of the traditional approach, which includes SentiCNN, MLT-ML4, and BERT-LLSTM-DL, which achieve an accuracy score of 0.90 to 0.95. Additionally, the comparison with large-scale foundation models (such as ChatGPT-4o and DeepSeek R1) in zero-shot prompt-based classification further validates the domain-adapted advantage of our

**Data availability statement:** All data used in this study are publicly available. The primary dataset, titled "Ancient Chinese Text (wenyanwen)", comprising classical Chinese literature texts, is openly accessible via the Kaggle platform and contains classical Chinese literary texts including poetry and prose suitable for natural language processing tasks such as sentiment analysis: https://github.com/jkxr58/Deep-Learning-Model-with-BERT-and-Graph-Attention-Networks. Kaggle Dataset Link. In addition, all data generated and analyzed during the study including tokenized input texts (translated into English), enriched embeddings, clustering labels, final centroids, and macro-averaged evaluation metrics (accuracy, precision, recall, and F1-score) are publicly hosted in a supplementary repository. Repository URL: https://github.com/jkxr58/Deep-Learning-Model-with-BERT-and-Graph-Attention-Networks. These materials include the minimal dataset necessary to replicate the study's findings, including all files used to generate tables, figures, and performance analyses. In continuation to this The sentiment lexicon used to enrich the embeddings was the NTUSD (National Taiwan University Sentiment Dictionary), which is also publicly accessible. No data access restrictions apply.

**Funding:** The author(s) received no specific funding for this work.

**Competing interests:** The authors have declared that no competing interests exist.

model in the classical Chinese text processing. These results demonstrate that the proposed model significantly enhances the handling of the intricate linguistic features and cultural nuances in classical Chinese texts, providing a robust solution for sentiment analysis in low-resource domains.

## 1 Introduction

In recent years, the rapid growth of technology has reshaped various fields, particularly in domains such as artificial intelligence (AI), machine learning (ML), and natural language processing (NLP). It has become increasingly important to understand public sentiments, as the world is now highly digitally interlinked. Sentiment analysis, a significant application in NLP, is crucial in uncovering the hidden feelings and thoughts associated with text [1]. This topic has been extensively researched across different languages, cultures, and situations, highlighting its growing relevance [2]. Since the world is increasingly interconnected and culturally diverse, analyzing sentiment in different languages is even more crucial [3]. Since a diversity of languages is a fundamental element of the global communication process, sentiment analysis methods must be developed to address the linguistic challenges associated with the process [4]. The sentiment analysis technique was developed from rule-based systems that entirely relied on expounded and clearly defined linguistic rules and patterns to identify sentiments in the text. However, with the advancement of ML techniques, this specific field has received several developmental boosts and seeded values that need to be considered. The ML is more flexible and generalizes the capability to deliver highly authentic and scalable sentiment analysis over wide-ranging textual data [5]. With the arrival of DL models, humor identification has changed significantly because of the capability of systems to identify complex syntactic and semantic patterns within text. Even so, modern neural models, such as convolutional neural networks (CNNs) and recurrent neural networks (RNNs), have proven capable of analyzing intricate linguistic patterns and shades of emotion. The essential approaches are beds for modeling relations between words and interpreting relations, making sense of tremendously advanced sentiment analysis systems [6]. Sentiment analysis based on deep learning is successfully used in various sectors of human activity, including social research, public policy, and commercial analytics. Namely, organisations employ it to identify their clients' issues, evaluate the overall attitude toward the brand they represent, and design adequate promotion campaigns [7]. The emergence of transformer-based models, such as BERT, has transformed NLP applications, enabling a more nuanced understanding of textual data. There have been investigations using BERT for sentiment analysis, highlighting its effectiveness in measuring contextual semantics. In addition, several researchers have optimized BERT for multimodal sentiment analysis, enhancing its ability to process and comprehend data from various sources [8]. Similarly, social researchers analyze the intensity and cycles of feelings within societies, and public politicians employ such techniques to determine how the public feels about particular matters of concern. These elaborate shows how deep learning can yield useful info from text [9].

Chinese classical literature has been a tremendous treasure trove of philosophy, culture, and feeling for thousands of years. It encompasses multiple genres, including poetry, prose, and historical records, that represent the evolution of Chinese thinking, art, and the behavioral code of nations. Indeed, great classics such as The Book of Songs (Shijing) and Records of the Grand Historian (Shiji) have not only imprinted themselves on literature but also occasioned pertinent philosophical thoughts and molded social orders throughout Chinese history. Over thousands of years, these texts provide access to the intricacies of human sentiment, shifts in society and lifestyle, and cultural norms that have influenced Chinese arts and ideas [10]. Examining the sentiments in these works can provide valuable insights into ancient China's emotional and sociocultural environments [9]. Sentiment analysis has become applicable due to advancements in NLP and the availability of Chinese literature digitised online. It also enables text analysis by time, genre, and dialect, and offers a new reading of the affective and evaluative dimensions of Chinese classical works [11]. These augment traditional qualitative approaches to facilitate scholars in recognizing transitions in literary thematics, those of culture, and variations in style [12]. Moreover, sentiment analysis in Chinese literature is crucial for the entire field of digital humanities, as it enables the large-scale study of literary development, authorship attribution, and stylistics. This enriches the previous historical and cultural analytical methods, allowing for the analysis of progressive changes in stylistic assignments and the evolution of authors' voices [13]. The researchers in [14] previously proposed a method for multi-task analysis using hierarchical attention mechanisms, which facilitated sentiment analysis of Chinese classical poetry. This model will be applied at the poem and line levels, respectively, since a style in a classical literary work can be either of these two. Such use was made of the hierarchical attention that assists in solving the fine-grained and emotion-overloaded correspondence inherent in poetry. The sentiment analysis was treated professionally.

Several studies have addressed the challenge of sentiment analysis in the context of literary works, particularly in languages with complex grammar and rich cultural backgrounds, such as Chinese. The work by [15] proposed a sentiment analysis model (SA-Model) using BERT-wwm-ext and ERNIE embeddings through hybrid word vectors. This approach employs multi-feature fusion to enhance the performance of semantic analysis of typical sections of classical Chinese texts and to address issues arising from the stylistic features of poems from distant periods. In the work [16], models of neural networks were used to study the poems of the Tang Dynasty, examining the linguistic and emotional features typical of ancient Chinese script. Some difficulties in analysing sentiment in this genre were described with an accent on the fact that emotions in poems written in ancient times are rather profound and diverse. Research carried out by [9] has presented a model termed BERT-LLSTMDL for coping with sentiment analysis of works of Chinese literature. This framework applies BERT for the most modern language representation, LSTM for sequence-to-capture, and DL for feature extraction, as it is challenging in this research area due to analysing Chinese literary texts' sentiment. To improve sentiment analysis results in traditional Chinese poetry, the author proposed enhancing sentiment analysis performance in traditional Chinese poetry. The framework uses sentiment labels for short lines and hierarchical attention to increase accuracy. With this method, sentiment analysis is done on both the poem and the individual lines simultaneously, and sentiment information from the short lines is used to enhance word- and sentence-level attention. This approach seeks to convey the subtle stylistic elements and emotional depth of ancient poetry more successfully.

Although development has been made, there are still several issues when dealing with sentiment analysis of Chinese classical literature because of its complex sentiment correlations and implicit meanings in the grammatical structure, metaphors, and allegories. Also, classical Chinese texts do not contain many direct words that can show the speaker's emotion. As such, earlier identification of sentiment scores is not always possible and entails using advanced models that can discern fine-grained emotional scripts inherent in the text, given the interdependent cultures and history. Further, both the conceptual metaphors and allusions add one more degree of indirection to the analysis, and the correct interpretation of these would require powerful approaches based on deep learning. Researchers must create models incorporating contextual understanding and fundamental word-based analysis to correctly infer feelings from these old literary works. A significant obstacle for supervised learning models, which depend on vast amounts of labelled data for efficient

training, is the dearth of annotated datasets for ancient Chinese literature. Because classical literature frequently doesn't have enough labelled instances for model training, this constraint makes it challenging to use traditional techniques for sentiment analysis in this field. LSTM networks, if used in sentiment analysis, have not remained without drawbacks, the key ones among which are the following: the training of non-local dependencies is considered extremely challenging, and the training computation demand is known to be much higher. These issues impact efficiency and, more so, large volumes of data, such as Chinese classical literature, because higher models are required for such texts [17]. In addition, the sequential nature of LSTM models hinders the models from benefiting from these contextual relations, as presented in these texts [18].

According to the abovementioned considerations, this study introduces a new unsupervised sentiment analysis model aimed at the Chinese classical literature. The proposed model fuses the strength of BERT-based contextual embeddings with graph attention networks (GATs) and sentiment lexicon enrichment to deal with the issues connected to analyzing ancient texts. This model deals with such issues as the absence of annotated datasets, the complexity of metaphors and implicit feelings, and the context dependencies of the classical Chinese language. The first important component of the model deals with extracting contextual word embeddings with the help of BERT-base-Chinese; hence, the model is capable of imbibing the semantic contents of words from their contexts. The second part involves incorporating sentiment-specific information of the NTUSD sentiment lexicon into the enrichment of embeddings so that the model can better comprehend the tone of emotional texts in classics. The third is the one that generates a graph-based representation in which words are transformed into nodes and employs GATs to determine their relationship in feature updates according to attention mechanisms. Lastly, the K-Means clustering classifies the sentiments into three categories. Positive, negative, and neutral. The findings from the experiments reveal that this unsupervised model outperforms the traditional approaches, with high accuracy, precision, recall and F1-scores, hence becoming a strong instrument to use in analyzing sentiments in the low-resource literary domains. This approach promotes sentiment analysis in classical literature and solves a scalable problem of studying other historical pieces rich in culture.

The key contributions of the paper are as follows:

- This paper introduces a new deep learning framework for Chinese classical literature sentiment analysis. It thus overcomes the usually encountered problem of the lack of labelled datasets in analyzing ancient texts.

- The model integrates sentiment-specific information from the NTUSD lexicon into the BERT-based embeddings, enhancing the model's ability to capture emotional nuances in classical Chinese texts.

- The paper provides Graph Attention Networks (GATs) to fill in the relationships between words regarding their co-occurrence and semantic relations.

- The proposed model is thoroughly examined using the experimental results that prove its better performance compared with the traditional sentiment analysis models, such as LSTM, BiLSTM, and CNN. The paper compares the proposed model to achieve better accuracy, precision, recall, and F1-scores. It demonstrates its effectiveness in sentiment classification of classical Chinese texts and provides a scalable solution for sentiment analysis in low-resource settings.

The rest of this research work is organized as follows: Section 2(Related Work) of the paper discusses the contribution of other researchers; Section 3(Materials and Methods) of the paper explains the framework of our proposed model; Section 4 (Performance Evaluation) of this work presents a performance evaluation of the proposed model and a discussion of the paper. Finally, the conclusion with the future research plan is given in Section 5(Conclusion and Future Work).

## 2 Related work

In the last few years, sentiment analysis has been one of the top techniques used in analyzing and interpreting Chinese literary works, as well as the emotions, themes and culture featured in the literary works [19,20]. The present literature

review aims to give a general idea of the research landscape, both about the methodology applied and the results obtained. In addition, it analyses the contribution of sentiment analysis to the understanding of emotions in the Chinese literary works, showing the complexity of relationships between themes, moods, and cultural settings in which the respective works were created. Computational methods aid in unveiling Chinese literature's hidden emotional and philosophical strata and understanding its sociocultural development.

Numerous previous studies have used sentiment lexicons, like lists of words and the subjective sentiment scores attached to such words, to obtain the overall sentiment score for any text based on the words in the text. These methods are quite easy to implement, have an interpretable nature, and have proven effective in domain-specific text analysis [21–25], and [26]. They perform especially well when processing short texts and delimited language, since the technology can be better used to detect specific linguistic shades. However, these approaches have their limitations when dealing with more complex texts, such as those found in the Chinese classical literature, where sentiment is usually implied rather than explicitly expressed by metaphors and backgrounds. Another method, which is based on labeled multilingual sentiment datasets, has been widely used for training supervised models of machine learning, such as Naive Bayes classifiers, Support Vector Machines (SVM), and Random Forest [27–31] and [32]. These techniques can be used for a medium-sized dataset and extract complex relations between data. However, relying on annotated data, the problem of feature selection, and difficulty in sarcasm interpretation are typically common and diminish their performance, especially when applied to massive literature or complex ones. Deep learning methods have caught up to overcome these limitations because no manual feature extraction is required, and complex patterns in text data can be automatically detected [33]. These include Convolutional Neural Networks (CNNs), Long Short-term Memory (LSTM) networks and transformer-based models such as BERT that managed to achieve better results in sentiment analysis due to picking up sophisticated emotional and syntactical patterns [34,35]. These models can be tuned by transfer learning approaches to make them capable of exploiting knowledge from source languages to better exploit the target language. For instance, BERT-based models have been used successfully to analyze the sentiments of Chinese news [36] and identify financial fraud from textual data [22]. These models' ability to capture subtle emotions and complex language structures has made them particularly effective for sentiment analysis in Chinese literary texts, especially when combined with fine-tuning strategies for specific domains.

In the significance of the Chinese classical literature, in recent studies, there has been an advent to practice using deep learning models to examine the complex emotional content of ancient texts. For example, a certain research presented a sentiment analysis model using the multidimensional knowledge attention to extract features belonging to the semantic and cultural context of ancient Chinese poetry [37]. Similarly, other studies have been using the Large Language Models (LLMs), for example, in sentiment analysis in classical Chinese literature, with a strong focus on Song Dynasty poetry (Song Ci) [38]. These studies show the possibility of the modern deep learning models working with classical Chinese language complexity and the possibility of conducting a better emotional content analysis. However, there are still needs to be met, especially in handling the problem of the absence of annotated datasets and the peculiarities of classical texts' language. Despite these obstacles, the further development of Chinese classical literature sentiment analysis techniques provides wonderful prospects for further development of understanding of the historical and cultural piquancy through computational methods.

## 3 Materials and methods

This section introduced a novel unsupervised sentiment analysis framework/model, considering adopting novel deep-learning techniques such as GATs and BERT. The input document represents the document that will be analyzed in the proposed deep learning-based sentiment analysis framework for Chinese classical literature, containing raw text. The first stage of the treatment process is preprocessing text, during which the given document is tokenized (divided into smaller parts, e.g., words or subwords). After tokenization, unwanted words in sentences are removed to concentrate on the main content, as special tokens are added to mark sentences and other vital parts.

The sequences are padded or cropped to a given uniform size to make the text consistent with deep learning models. Any preprocessing of the text further passes it through the BERT encoder, a model for pre-training that produces contextualised word embeddings. These embeddings reflect what each word means in the specific context, and since the complexities of classical Chinese literature depend heavily on context, this is essential. The embeddings are further perfected by including sentiment-specific information using a sentiment lexicon, such that the model can more accurately determine emotional tones within the text. Later, the embeddings are passed through Graph Attention Network (GAT). In this step, a graph is built, in which words are the nodes, and the connections between them are created assuming correlations and semantic dependencies between them. The attention mechanism of the GAT allows ranking the most essential relationships among words, contributing to improving the embeddings. Lastly, the K-means algorithm places the improved embeddings in groups, which divides the text's sentiment into three groups. Positive, negative, or neutral. This framework successfully overcomes the difficulties in the sentiment analysis of classical Chinese literature. It provides an insightful insight into the emotional and cultural aspects of the described literature and the proposed framework, as shown in Fig 1.

### 3.1 Dataset

The data set for this study is constructed from ancient Chinese text data collected to employ advanced natural language processing methods to predict linguistic tendencies and evaluate ancient Chinese classical works. The primary data

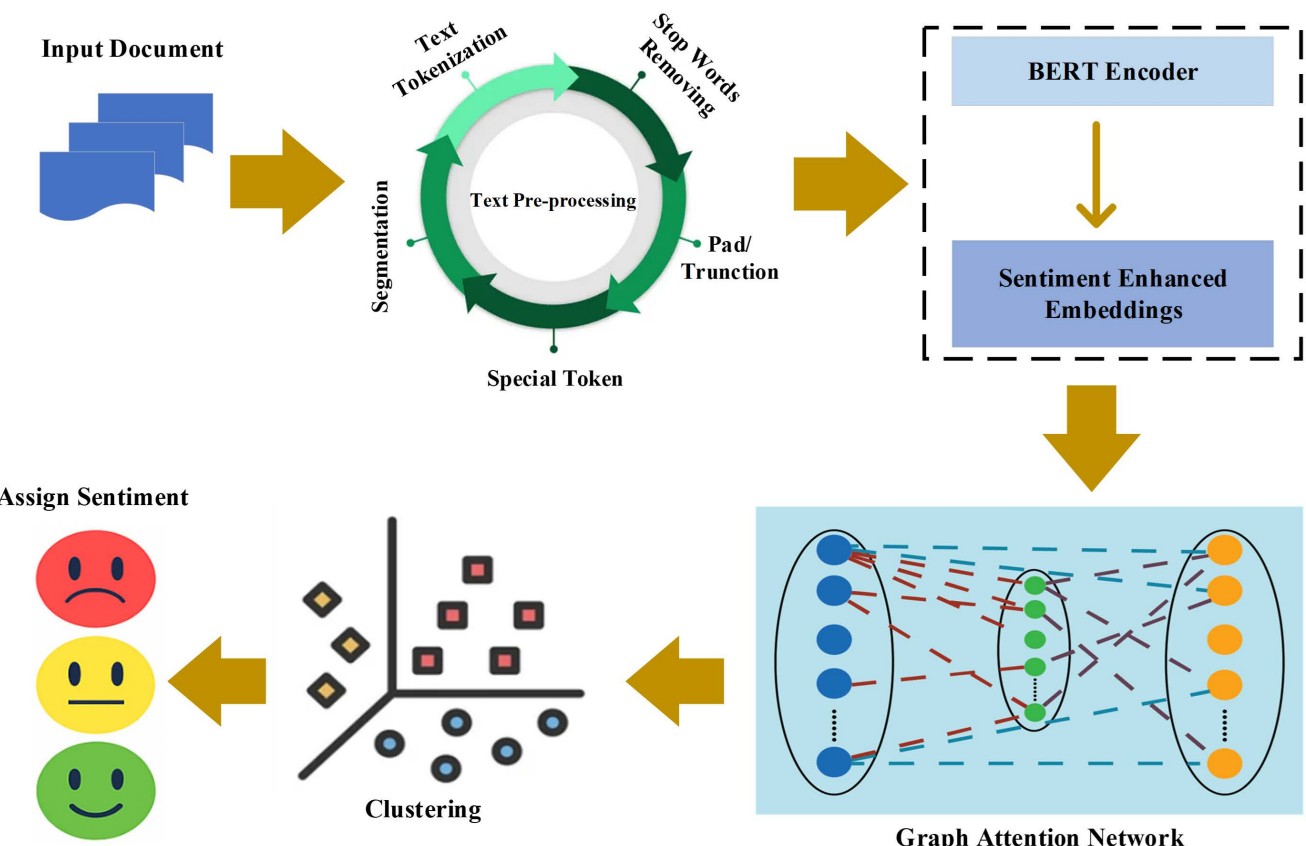

**Fig 1. Overall framework of the proposed model.**

source used is a Kaggle repository, which the authors mentioned in the work [10]. This dataset also comprises classical, bardic, and archaic poetry and prose, such as the Shijing, among the most credible historical sources and classical Chinese prose. These books accurately view conventional Chinese writings and societal norms and encompass historical records, philosophical commentaries, old King stories, and divinity personalities. No superior form of Chinese literature has captured historical, cultural, and ethical situations at this level more than classical literature. The dataset is available on the Kaggle open data repository at the Kaggle Dataset Link because the dataset covers various genres and emphasizes full text, it provides a comprehensive background for sentiment analysis. For the need for sentiment analysis and search for several themes within these works, these writings are ideal as they demonstrate features of linguistic difficulty and a syntactic density of an expression and metaphors.

### 3.2 Preprocessing

In preprocessing, complex raw Chinese classical text is transformed into an input format that can be well understood by deep learning models such as BERT and GAT. This stage deals with certain challenges that come with reading classical Chinese texts, such as rhetoric, carrying out hermeneutics, and styles of writing that withhold the extended meaning of a text [39]. For this pre-treatment, the text was prepared for subsequent contextual embedding extraction and graph-based modelling.

Tokenization splits the raw text into non-numeric, semantically related and usable tokens that are easier to interpret by deep learning models. This is why, for example, the WordPiece tokenizer works especially well for the BERT model, as it can process Chinese compound words, phrases, and even out-of-vocabulary words at once. It splits these into subword components, preserving semantic meaning even for previously unseen words. To tokenize process maps, each word. $w_i$, in a sentence to a tokenized representation $T(w_i)$:

$$T(w_i) = Tokenizer(w_i) \tag{1}$$

For the entire sentence $S = \{w_1, w_2, w_3 \ldots \ldots w_n\}$, the tokenized sequence becomes:

$$T(S) = Tokenizer(\{w_1, w_2, \ldots, w_n\}) \tag{2}$$

To ensure uniformity in sequence length ($L$), shorter sequences are padded with a unique token [$PAD$]. While longer sequences are truncated. This step is essential for batching inputs into fixed dimensions compatible with deep learning models.

$$S_{processed} = \begin{cases} S, \textit{If } S = L \\ Pad(S, L), \textit{If } S < L \\ Truncate(S, L), \textit{If } S > L \end{cases} \tag{3}$$

Special tokens are added to the sequences to denote specific roles as [$CLS$] Indicates the beginning of a sequence, providing a pooled embedding for classification. [$SEP$] Marks the end of a sequence or separates paired inputs.

$$S_{special} = \{[CLS]\} \cup S \cup \{[SEP]\} \tag{4}$$

Here, [$CLS$] and [$SEP$], are concatenated to the input sequence $S$.

This preprocessing pipeline ensures compatibility with the BERT-base-Chinese model and with GATs, paving the way for subsequent embedding extraction and sentiment classification.

**3.2.1 Contextualize embeddings extraction.** It could be suggested that contextualized embedding extraction can be considered an important stage of Chinese classical literature processing. After tokenization, padding and ADDINg special tokens, including *[CLS]* for classification and *[SEP]* as a separator, the resulting text is passed to the pre-trained BERT-base-Chinese model. This model produces fixed dimensions for each token, with the semantic meaning of the token contained within the entire context of the message. *BERT* has learned the syntax and semantics of the words. *BERT* provides context for each word through embedding, which is the essence of downstream tasks such as understanding classical Chinese literature. This process also helps in accurately determining the actual effect of a particular token, adjusting the representation of the token in the context of its neighbor tokens. For a token $t_i$ in the sequence:

$$S_{tokenized} = \{t_1, t_2, \ldots \ldots t_L\} : e_i = BERT(t_i, S_{tokenized})$$

(5)

Where $e_i$ is the contextualized embedding for $t_i$. One of the last hidden layers of the BERT-base-Chinese model is the source of these embeddings. The embeddings capture the subtleties of ancient Chinese sentence construction. $e_i$, which are constantly impacted by surrounding tokens. The BERT-base-Chinese model, optimized for modern and classical Chinese, guarantees that the embeddings are suitable for handling intricate idioms and expressions. Fig 2 displays the BERT encoder for contextualized embeddings.

The output from this step is a matrix $E \in R^{LxD}$, where $L$ is the sequence length and d is the embedding dimensionality. These embeddings serve as the foundation for subsequent enhancements and graph-based sentiment analysis.

## 3.3 Sentiment enriched embeddings

Enhance the *BERT* embeddings by integrating sentiment-specific information from an external *NTUSD* sentiment lexicon. This step tailors the embeddings to better reflect sentiment patterns relevant to Chinese classical literature. The *NTUSD* lexicon is used to assign sentiment scores $s_i$ to each token. $t_i$ Based on its presence and polarity in the lexicon. If $t_i$ is not in the lexicon, $s_i = 0$.

$$s_i = \begin{cases} NTUSD(t_i), if\ t_i \in Lexicon \\ 0, otherwise \end{cases}$$

(6)

Combine the sentiment scores $s_i$ with the *BERT* embeddings $e_i$, to generate enriched embeddings $e_i^s$ A simple concatenation or weighted addition can be used:

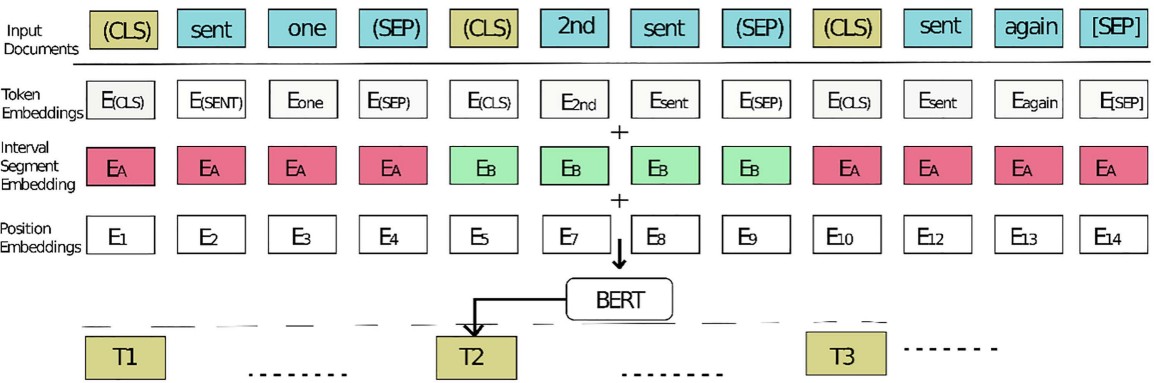

**Fig 2. BERT Encoder for contextualised embeddings.**

$$e_i{}^{(s)} = Concat\,(e_i, si) \tag{7}$$

or

$$e_i{}^{(s)} = e_i + \alpha si \tag{8}$$

Here in Eq. 8, $\alpha$ *is* a scaling factor. The output of this step is $E(s) \in RL \times (d+1)$ encapsulates contextual and sentiment-specific information. These embeddings are then used to construct a relationship graph in the next stage.

## 3.4 Graph attention networks

Graph Attention Networks (GATs) are one of the types of neural networks designed to process graph-structured data. As opposed to the traditional convolutional neural networks (CNNs) that are intended for grid-like data (e.g., images), GATs specialize in working with non-Euclidean, irregular graph structures. In a graph, data points (nodes) are connected with edges where the relationships are, and GATs use these relationships to update node representation. The central element of GATs is the attention mechanism, which enables the model to have different weights for the neighboring nodes to aggregate their information. Rather than treating all the neighbors equally, GATs use the attention coefficients where the most important neighbors are emphasized so that the model can learn the contextual importance of different nodes. This self-attention mechanism allows GATs to effectively deal with dynamic and complex graph structures. GATs are used for sentiment analysis, especially for tasks related to text or language, to model dependencies of words or phrases as nodes in the graph. The attention mechanism allows the model to concentrate on contextually important word associations, such as co-occurrence or syntactic connections, essential in intentional and semantic variety detection in the text. By adapting to these weighted relationships of nodes, GATs significantly improve the model to capture complex contextual information. Fig 3 illustrates the architecture of the Graph Attention Network (GAT), highlighting the process of updating node representations based on the attention mechanism applied to neighboring nodes in the graph.

**3.4.1 Graph layer.** In the proposed unsupervised sentiment analysis methodology, the graph construction step plays a critical role in modelling the relationships between words and their context and in facilitating clustering based on the semantic and syntactic dependencies of the words. In this step, we represent words as nodes and define edges based on their relationships, forming a graph $G = (V, E)$, where $V$ *is* the set of nodes (words) and $E$ is the set of edges, respectively.

- **Representing Words as Nodes:** Each token $t_i$ from the sentiment-enriched embedding matrix $E^{(s)}$ is treated as a node $v_i. V = \{v_1, v_2, \ldots\ldots, v_L\}$. Where is the number of tokens in the sequence. The feature vector of each node is initialized using its sentiment-enriched embedding. $e_i{}^{(s)}$.

- **Edges Creation among Nodes:** Edges are created between nodes based on two criteria: Co-occurrence within a sentence, in which two tokens occur. $t_i$ and $t_j$ When co-occur in the same sentence, an edge is created:

$$E_{(co-occurence)} = \{(v_i, v_j)|t_i, t_j \in samesentence\} \tag{9}$$

The second is the Cosine Similarity of Word Embeddings, in which Edge weights are computed using cosine similarity between embeddings. $e_i{}^{(s)}$ and $e_j{}^{(s)}$.

$$w_{ij} = cosine_{sim\left(e_i{}^{(s)}, e_j{}^{(s)}\right)} = \frac{e_i{}^{(s)}.e_j{}^{(s)}}{|e_i{}^{(s)}||e_j{}^{(s)}||} \tag{10}$$

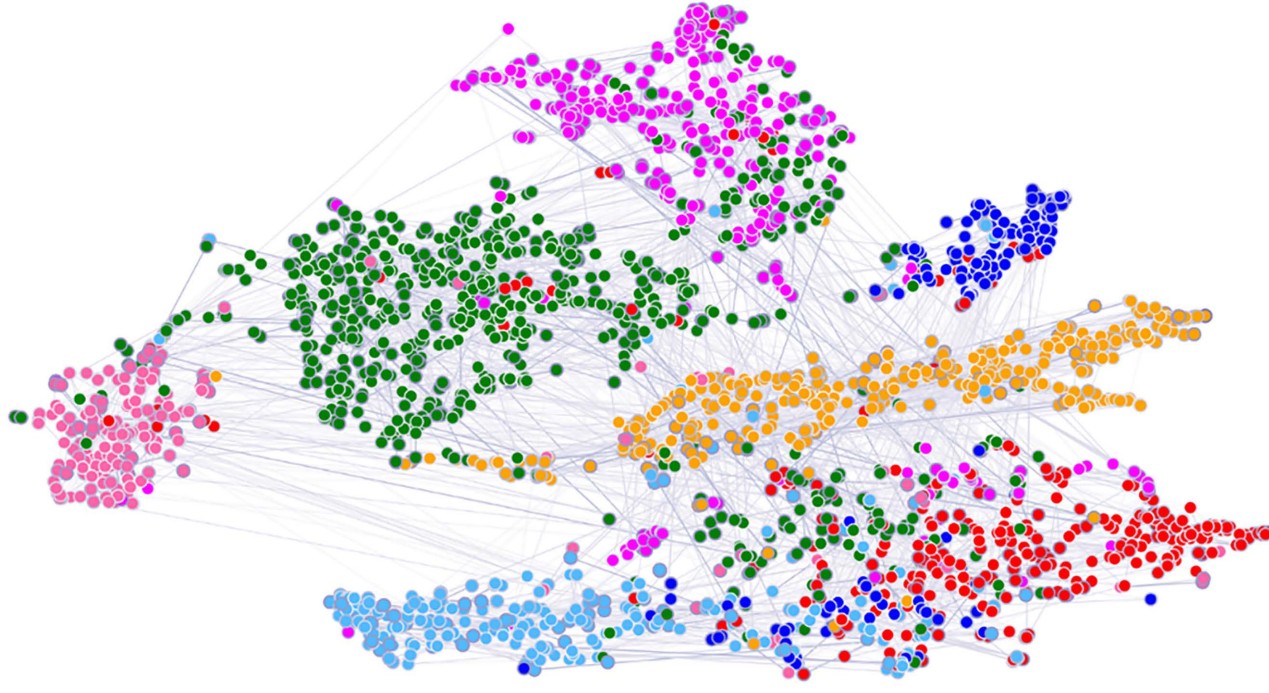

**Fig 3. Architecture of the GAT model.**

At the edge, the weight is calculated using Eq. 11 below:

$$w_{ij} = \alpha E_{co-occurence} + \beta cosine_{sim\left(e_i^{(s)}, e_j^{(s)}\right)}$$

(11)

- **Graph Representation:** The resulting graph is represented as:

Adjacency Matrix A: Encodes edges, where $A[i][j] = w_{ij}$, if an edge exists, otherwise 0.

Feature Matrix X: Encodes node features derived from $E^{(s)}$.

### 3.5 Graph attention networks processing

GAT processes the constructed $G = (V, E)$ Update node features by aggregating information from neighbouring nodes. Each node $v_i$ attends to its neighbours $v_j \in N(i)$. The Attention score $\alpha_{ij}$, is computed as:

$$\alpha_{ij} = \frac{exp(Leaky\,Re\,LU\left(a^T\left[We_i^{(s)}||We_j^{(s)}\right]\right))}{\sum k \in N(i)exp(Leaky\,Re\,LU\left(a^T\left[We_i^{(s)}||We_j^{(s)}\right]\right))}$$

(12)

Here above, Eq. 12 $W$ is the weight matrix for linear transformation, $\alpha$ is the Attention weight vector, and $W$ is the concatenation operation. To update features $h^{i'}$, are computed as a weighted sum of the neighbours' features:

$$h^{i'} = \sigma\left(\sum j \in N(i)\alpha_{ij}We_j^{(s)}\right)$$

(13)

Above, Eq. 13 $\sigma$ is an activation function (e.g., ReLU). The output is an updated feature matrix $H \in R^{Lxd'}$ is the dimensionality of the updated embeddings.

Sentiment analysis has become useful for getting emotional and topical inferences from text, particularly for intricate literary works. Chinese classical literature, which is characterised by numerous metaphors, cultural references, and historical perceptions, provides a unique challenge for sentiment analysis. The suggested algorithm uses state-of-the-art methods like BERT embeddings, Graph Attention Networks (GAT), and K-means clustering to overcome these issues. It provides a working solution for analyzing classical texts' emotional depth. Combined, these methods allow the framework to provide a more fine-grained sentiment understanding in literary works even without large datasets for annotation.

```
Algorithm 1. Sentiment Analysis using GAT.
```
**Input:**
```
• Raw text document
```
$T = \{t_1, t_1, \dots, t_n\}$ `, where` $t_i$ `represents the i`ᵗʰ `word in the text.`
**Output:**
```
• Sentiment labels
```
$y_t \in \{positive, negative, neutral\}$`for each document` $T$.
 ***Step 1:***`Preprocessing`
 `• Tokenize text, remove stop words, and add special tokens.`
 `• Pad or truncate sequences to a fixed length` $L$.
 ***Step 2:***`Sentiment Lexicon Enrichment`
 `• Retrieve sentiment score` $s_i$, `from the NTUSD lexicon and concatenate with BERT embeddings to form enriched embeddings` $E_{enriched}(w_i) \in \mathbb{R}^{d+1}$.
 ***Step 3:***`Graph Construction`
 `• Represent words as nodes and construct edges based on co-occurrence and cosine similarity of enriched embeddings.`
 ***Step 4:***`Graph Attention Network (GAT) Processing`
 `• Compute attention scores` $\alpha_{ij}$ `for neighboring nodes and update node representations` $H_i$ `by aggregating weighted features:`

$$H_i = \sigma \left( \sum_{j \in N(i)} \alpha i_j W E_{enriched}(w_j) \right)$$

 ***Step 5:***`Clustering`
 `• Apply K-means clustering on updated embeddings` $H$ `to categorize sentiment (positive, negative, or neutral).`
 ***Step 6:***`Output`
 `• Return sentiment labels based on assigned cluster centroids.`

## 3.6 K-means clustering and sentiment labels assignment

The organization of data, as well as data pattern recognition, is boosted by the process of K-means clustering that categorises data points into clusters based on how close they are to the centroids. The before and after the application of the k-means clustering algorithm. They appear on the left as a pile of black dots, i.e., a haphazard and scattered formation with no clusters. The data has no natural structure or form and has not been sorted into humanly meaningful clusters. After K-means clustering (shown on the right side), the data points are sorted into clusters using different colored ellipses. Two clusters are created here, the first in cyan and the second in red. The clusters are being found based on how close the given points are to the clusters' centroid, the center point of the given points within a particular group. The algorithm maps every data point to the closest centroid, and in doing so, the algorithm divides the points into groups. The workings of K-means clustering involve iteration until the within-cluster variance is minimized through shifting the centroids. At first, the centroids are picked randomly, and the points are assigned to the closest centroid. The centroids are computed again using the new points, and the process is carried out until convergence, when the centroids do not move substantially. As a

confusing technique, this clustering procedure is highly employed in segmentation, pattern recognition, and classification of problems. Fig 4 shows the K-means clustering process as data points are divided into unique sentiment types (positive, negative, or neutral) according to their proximity to the centroids of the clusters.

To classify sentiment, unsupervised clustering, k-means, is applied to the final node representations $H$ output generated from GATS. Clustering partitions the nodes into $k$ clusters, each corresponding to a sentiment class (positive, negative, neutral).

Cluster centroids are mapped to sentiment labels based on proximity to predefined sentiment vectors or a manual evaluation of cluster compositions. Let $C = [C_1, C_2, \ldots\ldots, C_k]$, represent the clusters, and $c_i$, denote the centroid of the cluster $C_i$. The label $l_i$ for $C_i$ is assigned based on:

$$l_i = argmax_s \ cosin_{sim(c_i, v_s)} \tag{14}$$

## 4 Performance evaluation

In this research, we proposed an unsupervised sentiment analysis model tailored to Chinese classical literature, leveraging advanced deep learning techniques such as the BERT model and GATs. The approach addresses the inherent challenges of analyzing texts characterized by complex linguistic structures, context-rich semantics, and cultural nuances, which are prevalent in classical Chinese works. By eliminating the reliance on labeled datasets, the model provides a scalable and effective solution for sentiment analysis in specialized domains like historical and philosophical writings, ancient poetry, and classical fiction. Our method enables fine-grained sentiment classification into positive, negative, and neutral classes by using sentiment-augmented graph embeddings in conjunction with contextualized embeddings. This evaluation chapter demonstrates the versatility and effectiveness of the chosen model to detect affective attitudes and work with the tendencies in ethical themes in ancient Chinese writings by explaining how the model performed in the experiment, comparing the sentiment distribution in various literary genres, and assessing the model's generality and expandability. Furthermore, we present detailed information about the implementation settings and hyperparameters applied in this work.

To measure the effectiveness of the sentiment model designed earlier, the authors rely on the following key performance indicators: error rate, precision, sensitivity, and F-measure. They help determine the model's effectiveness in classifying data into various types [28]. A brief explanation of each of the employed metrics is given below:

**Accuracy:** The measure of the model's quality is determined by the number of properly classified instances over those present in the data set. This tells the extent of the success of the model.

$$Accuracy = \frac{\sum_{c=1}^{N} TPc}{\sum_{c=1}^{N} (TPc + FPc + FNc)} \tag{15}$$

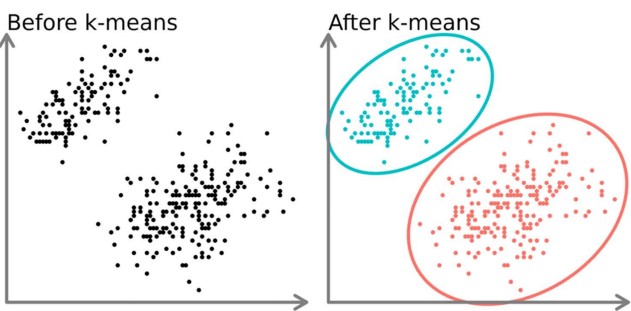

**Fig 4. K-means clustering processing.**

**Precision:** the measure of the fraction of actual positive instances expected in a set of positive instances predicted by the model. This feature is essential, especially when it is required to reduce false positives.

$$precision_{macro} = \frac{1}{N}\sum_{c=1}^{N} \frac{TPc}{TPc + FPc} \tag{16}$$

**Recall:** The measure of the percentage of instances of a specific relevant sentiment expected to be classified correctly by the model. This is very important when reducing the number of desired false negatives.

$$recall_{macro} = \frac{1}{N}\sum_{c=1}^{N} \frac{TPc}{TPc + FNc} \tag{17}$$

**F1-Score:** It has the measure, calculated using the harmonic mean of precision and Recall measures. Unlike the others, this measure does not depend on data distribution or abnormalities in data.

$$F1_{macro} = \frac{1}{N}\sum_{c=1}^{N} 2 \times \frac{precision_c \times recall_c}{precision_c + recall_c} \tag{18}$$

In the above equations, TPc true positives for class c, FPc false positives for class, FNc false negatives for class c, TNc: true negatives for class c, and N is the total number of sentiment classes $N = 3$. These metrics will form a structure that will independently assess the strengths, weaknesses, and efficiency of leveraging these complexities to annotate classical Chinese literature. The proposed framework is, therefore, made more reliable and scalable by this comprehensive assessment approach. The evaluation of sentiment classification is a macro-averaged three-class arrangement, in which the predicted sentiment categories, positive, negative, and neutral, are accorded equal weight. Precision, recall, and F1-scores are determined for every class, and then an average is taken to get the final scores. This technique does not suffer from the bias caused by class imbalance and provides unbiased evaluation on the part of all sentiment types. No binary classification or simplification of sentiment classes was carried out during evaluation.

## 4.1 Implementation settings

The proposed unsupervised Chinese classical literature sentiment analysis framework requires thoroughly chosen hyperparameters and a clearly defined implementation plan to report optimal outcomes. BERT-base-Chinese tokenizer is employed for vocabulary segmentation based on a pre-trained subword tokens lexicon. The sequence length is fixed, and padding is added where required to ensure consistent input shapes over batches. A batch size of 32 was chosen to optimise GPU memory consumption and stability of gradients, allowing for efficient training while respecting the computational cost. The BERT-base-Chinese model outputs the contextualised word embeddings of dimensionality 768, wherein understanding deep semantic and syntactic attributes of the given token in the context is represented. These embeddings are also enhanced with sentiment-specific scores from the NTUSD sentiment lexicon, leading to an overall 769 embedding dimension per token. This augmentation improves the model's representation of subtle emotional tones in literary texts. The graph representation emerges from these enriched embeddings, where nodes represent words while edges are established through sentence co-occurrence, cosine similarity of embeddings, and syntactic dependencies. The graph is processed via a GAT, consisting of two layers with eight attention heads each. Each attention head projects to a subspace of 64 dimensions, resulting in final node embeddings of 512 per node, capturing updated node representations. These layers learn to semantically and syntactically prioritise and dynamically aggregate neighbouring nodes' information.

From the computational perspective, approximately 1.2 million trainable parameters are involved within the GAT component, whereas the BERT-base-Chinese model accounts for about 110 million parameters. Thus, the total model complexity approaches approximately 111.2 million parameters when BERT is fine-tuned, and 1.2 million parameters when

BERT is frozen. This configuration strikes a compelling balance between performance and scalability, which qualifies it for massive scaling of text processing workloads. Finally, K-Means clustering executes the ultimate sentiment classification stage through its unsupervised approach, which operates without trainable parameters. Node representations undergo clustering into three distinct sentiment categories: positive, negative, and neutral, based on a k-value of 3. This step aligns with the emotional distribution expected in analysing classical literary texts. To avoid overfitting, between layers, a dropout rate of 0.3 is used, randomly turning off neurons during training, to improve generalisation. Optimisation is done using the Adam optimiser, and a dynamic learning rate scheduler ensures smooth convergence with learning rate $= 2 \times 10^{-5}$. The model is trained for 10 epochs using early stopping concerning validation loss to avoid overtraining.

## 4.2 System requirements and specs

The training and evaluation of the proposed unsupervised sentiment analysis framework were conducted on a high-performance computational system to ensure efficiency and accuracy. The detailed system specifications are presented in Table 1 below:

## 4.3 Models for comparison

In this section, we compare our proposed model to the state of the art in Sentiment Analysis presented in the different domains of Sentiment Analysis literature. To compare our framework with modern deep learning architectures and evaluate their effectiveness in this comparison. SentiCNN [40], Sentiment Convolutional Neural Network, evaluates the sentiments of the sentences by using contextual and sentiment-specific data with the help of the above architecture. The semantic relations between the words, including the contextual analysis, are captured by word embedding, while other lexicon data are obtained from standardized lexicons. With these two methods, SentiCNN can explore the word context and sentiment cues in detail, thus having higher sentiment classification rates. The accuracy, precision, recall, and F-Score of the proposed SentiCNN model are 0.90, 0.89, 0.92 and 0.90. MLT-ML$_4$ [41] shows that the multi-task learning framework is associated with the word-level latent topic distributions used in the topic model and the word-level attention vectors used in the sentiment classifier. A combination of sentiment categorization and topic modeling is made possible by mutual learning in the process distribution. The framework incorporates deep learning techniques, a Neural Topic Model for facilitating the modelling of topics, and Recurrent Neural Networks for handling time series data. The values of accuracy, precision, recall, and F- Score obtained in this study are 0.79, 0.81, 0.83, and 0.78, respectively.

**Table 1. System specification and requirements.**

| Component | Details |
|---|---|
| **Hardware Environment** | |
| System Model | Dell Precision 7820 Tower |
| Processor | Intel Xeon W-2295, 3.0 GHz |
| Graphics Processing Unit (GPU) | NVIDIA GeForce RTX 3090 |
| RAM (Random Access Memory) | 128 GB DDR4 |
| Storage (ROM) | 2 TB NVMe SSD |
| Operating System | Windows 11 |
| **Software Environment** | |
| Python Version | 3.9 |
| Platform | Google Colab-Pro |
| Frameworks | TensorFlow 2.11 or PyTorch 1.13 |
| CUDA Toolkit Version | 11.7 |
| Other Libraries | Transformers, Scikit-learn |

B-MLCNN [42] is a deep-learning approach for document-level sentiment analysis. This method endeavoured to enhance the precision and understanding of sentiment analysis in larger textual contexts by integrating up-to-date deep learning prescriptions, such as transformer models and recurrent neural networks. The present study's accuracy, precision, recall, and F-Score are 0.95, 0.88, 0.95, and 0.95, respectively. T-Caps [43] worked on the information loss problem and proposed a sentiment categorization using a capsule network model. The model employs the Transformer to capture shallow text attributes to guarantee realistic primary feature extraction. The method associates local textual features with holistic emotion cumulative favour/opposition through the capsule network, global parameter sharing and optimal dynamic routing update procedures. Statistical measures of this study are accuracy of 0.94, precision of 0.93, recall of 0.95, and F-Score of 0.93. KGAN [44] is a knowledge graph augmented network that records sentiment feature representations from several angles, including knowledge-based, context-based, and syntax-based. KGAN first learns the syntactic and contextual representations to fully extract the semantic characteristics. After KGAN incorporates the knowledge graphs into the embedding space, an attention mechanism is used to extract the aspect-specific knowledge representations further. This work has an accuracy of 0.84, a precision of 0.82, a recall of 0.87, and an F-Score of 0.78. BERT-LLSTM-DL [9] is a state-of-the-art deep learning system designed for sentiment analysis of Chinese literature. Boundary loss works out the correct features by incorporating advanced deep learning algorithms, LSTM networks in sequential data analysis, and additional BERT for enhanced language representation. This work has an accuracy of 0.95, a precision of 0.96, a recall of 0.94, and an F-Score of 0.95. ChatGLM-6B [38], emphasising Song Dynasty poetry (Song Ci), refined the LLaMA 2 and Qwen LLMs for sentiment analysis on traditional Chinese literature. These fine-tuning techniques aimed to more precisely traverse and grasp Song Ci's complex language and emotional content. Both supervised methods and reinforcement learning from user feedback are used in the fine-tuning process, which is especially intended to match the models with Song Ci's historical and cultural background. This work has an accuracy of 0.91, a precision of 0.89, a recall of 0.92, and an F-Score of 0.84. Proposed Model, A specially developed unsupervised Chinese classical literature sentiment analysis model with state-of-the-art deep learning techniques such as the BERT model and GATs. The proposed method addresses challenges in analyzing texts where semantics and pragmatics involve socio-contextualized, non-linear, and layered concepts distinctive from the classical Chinese language. The approach represents a cost-effective solution to the problem of abstract domains, including classical fiction, ancient poetry, and philosophical and historical writings, thereby reducing the need for labeled datasets. The results of the proposed study includes an accuracy of 0.95, precision of 0.97, recall of 0.96, and F-Score of 0.91, respectively.

## 4.4 Overall performance of the model

The suggested sentiment analysis model, which has been developed with the help of GATs for comprehending intricate connections between words and BERT for contextualized word representations, is analysed in this section. A literature area characterized by both dense linguistic and thematic context, Classical Chinese literature, was employed to evaluate the model. The outcomes demonstrate how well the model correctly divides attitudes into three main groups: neutral, negative, and positive. These problems and patients' experiences can be categorized into three categories, consisting of neutral, negative, and positive. The results of the suggested model for multiple iterations are presented in Table 2 below. First, the accuracy of the proposed model is increasing and exceeds 0.97, which means that it can control the nuances of classical Chinese literature. The high accuracy exhibited by the model shows its accuracy in sentiment classification, even for the most syntactic density that accompanies literary and historical writings. One of the most significant advantages of the proposed model is that it carries out the basic function of combining BERT and GATs to classify feelings successfully. Regarding measurement precision, the model delivers average predictability rating values of 0.96 to 0.98. The following results show how much the model reduces false positives and accurately identifies relevant sentiment categories. Sustaining such a high degree of accuracy is crucial to ensure that appropriate sentiment classifications that align with current

**Table 2. Results of proposed model on various iterations.**

| Iteration | Accuracy | Precision | Recall | F1-score |
|---|---|---|---|---|
| Iteration 1 | 0.94 | 0.96 | 0.95 | 0.91 |
| Iteration 2 | 0.94 | 0.96 | 0.95 | 0.90 |
| Iteration 3 | 0.95 | 0.97 | 0.96 | 0.91 |
| Iteration 4 | 0.96 | 0.97 | 0.96 | 0.91 |
| Iteration 5 | 0.97 | 0.98 | 0.98 | 0.89 |

requirements are achieved in traditional Chinese literature's strong historical and thematic undertones. Likewise, the recall scores range from 0.95 to 0.98, which indicates that the model can identify a large fraction of truly positive thinking.

This section of the model's function is crucial, given that it also reveals the ability to identify sentiment in less obvious cases, which is challenging to identify in large text analyses when working with complex structures of literary genres. These F1 scores, which involve the fine equilibrium between precipitation and recall that consistently remains above 0.91, underpin the model's efficiency and reliability. This also demonstrates that the model is accurate for the purpose and capable of handling the relevant recall/precision compromise, so the system is properly set to get large amounts of relevant and pertinent sentiment data. The same performance testing is done to get more insights into the performance of our suggested model, as done by the research team [5]. The decision to use text sentiment analysis to assess the various Chinese literary works has been presented in Table 3. Rows in the given table are specific to the particular sentence; the sentiment scores are positive, negative, and neutral. The table also shows the projected sentiment associated with the category that scored the highest out of the three. A few significant patterns become apparent while looking over the sentiment results. For instance, texts like "这本书是非常有趣" (This book is fascinating.) and "这个产品的质量非常好" (The quality of this product is very good.) received strong positive sentiment scores, leading to the prediction of a positive sentiment. On the other hand, texts such as "这个电影太令人失望了" (This movie is too disappointing) and "这次旅行经历真是太糟糕了" (This travel experience is terrible) scored highly on the negative sentiment scale, resulting in pessimistic predictions. Additionally, some texts reflect a balance of sentiments, which led to neutral sentiment predictions. For example, "这个餐厅的食物味道很好, 但服务很慢" (The food in this restaurant is good, but the service is slow.) and "我喜欢这个城市的风景, 但交通拥堵" (I like the scenery of this city, but the traffic is congested) received similar scores across positive, negative, and neutral categories, indicating a mixed or neutral sentiment. In light of the preceding results, the model's applicability in categorising the sentiments in pre-modern Chinese works is evident from the analysis of contextual meanings, whereby the texts are sorted according to the positive, negative, or neutral emotions conveyed.

**Table 3. Proposed model scores and sentiment prediction.**

| Chinese Text | Positive Score | Negative score | Neutral Score | Predicted/Assigned Sentiment |
|---|---|---|---|---|
| 这本书是非常有趣的 | 0.97 | 0.01 | 0.06 | Positive |
| 这个餐厅的食物味道很好, 但服务很慢 | 0.21 | 0.33 | 0.45 | Neutral |
| 这个电影太令人失望了 | 0.04 | 0.89 | 0.21 | Negative |
| 这个产品的质量非常好 | 0.91 | 0.02 | 0.31 | Positive |
| 这次旅行经历真是太糟糕了 | 0.03 | 0.74 | 0.21 | Negative |
| 这个软件的用户界面太复杂了 | 0.01 | 0.98 | 0.03 | Negative |
| 我对这个服务感到非常满意 | 0.98 | 0.08 | 0.21 | Positive |
| 我喜欢这个城市的风景, 但交通拥堵 | 0.31 | 0.01 | 0.84 | Neutral |
| 这个演员的表演非常精彩, 但剧情太简单了 | 0.32 | 0.12 | 0.65 | Neutral |
| 我觉得这个活动很有意思 | 0.85 | 0.01 | 0.32 | Positive |

Table 2 compares the suggested model with the state-of-the-art deep learning models for sentiment analysis. Over the years, our technique outperformed the baseline models in all measures used for evaluation, namely accuracy, precision, recall, and F1 score. Specifically, this extensive improvement demonstrates how effectively adding BERT embeddings, GATs, and unsupervised clustering adaptively preserves intricate syntactic/semantic features of traditional Chinese literature. The ability of the proposed model to optimize the deep learning requirements indicates that the model is adaptive to certain challenges posed by classical literature, including difficult language structures and hidden sentiment signification. For one, the model could be more precise at sentiment classification even when there are very limited amounts of annotated data, thanks to the incorporation of graph-based representation learning and domain-specific sentiment lexica.

While this framework presents deep analytic results for both thematic and emotional features of Chinese classical literature, the sentiment analysis part provides insights. Finding cultural moods for texts of various epochs, historical documents, and philosophical and poetic works will help reveal intellectual and poetic moods more thoroughly. On this premise, a better understanding and probing of sentiment in various literary settings becomes possible with potentially real-world applications in literary criticism, history, and culture. It eliminates the problem of a lack of labelled data sets characteristic of many specific disciplines, such as historical Chinese literature. This scalability and flexibility of the method make it especially suitable for other domains in which only a small portion of the data is labelled. It is also interesting to note that integrating BERT and Graph Attention Networks (GATs) works well. BERT achieves sophistication by using contextual embeddings with syntactic and semantic nuances; conversely, GATs add value to the model in analyzing essential linkages between the words and relations between texts. This interaction improves the model's ability to identify sentiment within the more complex structures of classical literature. The performance indicators calculated in model accuracy, precision, recall, and the F1 score show how useful the model is. This makes it capable of learning from extensive data without getting too fixated on some stereotyped results, which makes it highly robust. These results show that applying the proposed framework to a classical Chinese literature text corpus and extending it to another textual dataset enlightens the natural language processing study.

The performance sets the contrast of the proposed sentiment analysis model with a series of state-of-the-art models based on top metrics: accuracy, precision, recall, and F1-score. All the scores are macro-averaged across three sentiment categories, i.e., negative, neutral, and positive. The proposed model is better than several comparison models, gaining an accuracy of 0.95, a precision of 0.97, a recall of 0.96, and an F1-score of 0.91. It is at least as good as models such as BERT-LLSTM-DL and B-MLCNN, which delivered similar accuracy and recall with lower precision. The proposed model outperforms MLT-ML4, which have lower metric scores. These findings are shown in Table 4, and the model proves useful in sentiment classification in complex textual data, particularly in classical literature. It should be noted that all scores are macro-averaged over the three sentiment categories (negative, neutral, and positive). No binary simplification was used.

We also performed a direct empirical comparison with two state-of-the-art foundation models, ChatGPT-4o and DeepSeek R1, in a zero-shot learning environment to thoroughly evaluate the efficacy of our suggested unsupervised sentiment analysis framework. The same input dataset of literary sentences in classical Chinese (with translations where necessary) and a uniform instruction format were given to each model.

**"Classify the following Chinese literary text's sentiment as either positive, negative, or neutral."**

Following that, the same performance metrics used in our study Accuracy, Precision, Recall, and F1-score were used to assess the predicted sentiments from ChatGPT-4o and DeepSeek R1. Table 5 presents a summary of the evaluation results. With an F1-score of 0.91, the suggested model continuously outperforms these powerful general-purpose models on all metrics, while ChatGPT-4o and DeepSeek R1 achieve 0.90 and 0.88, F1-scores respectively.

These results imply that although foundation models such as ChatGPT-4o and DeepSeek R1 show remarkable generalization abilities even in zero-shot sentiment classification, they might still have limitations when used on linguistically

**Table 4. Comparison of the proposed model with existing models.**

| Existing Models | Accuracy | Precision | Recall | F1-score |
|---|---|---|---|---|
| SentiCNN [40] | 0.90 | 0.89 | 0.92 | 0.90 |
| MLT-ML$_4$ [41] | 0.79 | 0.81 | 0.83 | 0.78 |
| B-MLCNN [42] | 0.95 | 0.88 | 0.95 | 0.95 |
| BERT-LLSTM-DL [9] | 0.95 | 0.96 | 0.94 | 0.95 |
| **Proposed Model** | 0.95 | 0.97 | 0.96 | 0.91 |

**Table 5. Comparison of the proposed model with ChatGPT-4o and DeepSeek R1.**

| Models | Accuracy | Precision | Recall | F1-score |
|---|---|---|---|---|
| **Proposed Model** | 0.95 | 0.97 | 0.96 | 0.91 |
| **ChatGPT-4o** | 0.90 | 0.89 | 0.91 | 0.90 |
| **DeepSeek R1** | 0.88 | 0.87 | 0.89 | 0.88 |

complex, domain-specific corpora like Chinese classical literature. It can be difficult to decipher these texts without specific fine-tuning or domain adaptation because they frequently rely heavily on idioms, metaphorical structures, historical allusions, and philosophical allusions. In contrast our framework is better design to better align with the intricate stylistic and emotional characteristics of classical Chinese literature especially in low-resource and label-scarce scenarios where LLMs often fall short unless specifically adapted or prompted with domain knowledge.

However, given the exceptional adaptability and flexibility of contemporary LLMs, we think our strategy can be complementary rather than competitive. Future research will focus on hybrid approaches that combine domain-specific modeling using lexicons and graph architectures with general-purpose reasoning from LLMs, utilizing the best features of both paradigms.

### 4.5  Generalization and scalability of proposed model

The proposed unsupervised sentiment analysis model has better data generality and expansiveness in textual styles. It successfully handles large amounts of data, such as large and intricate data structures, including extended collections of ancient Chinese literature, with no decline in efficiency by using BERT embeddings and Graph Attention Networks. This scalability is important mainly for processing large bodies of text to investigate inter-period, inter-genre, and inter-style consistency. This guided nature of the model is made even more effective by the ability of the program to function in an unsupervised way and, thus, to successfully apply a large number of types of ancient Chinese literature that include fiction, poetry, history, and philosophies of China. This versatility supports the analysis of sentiment and other temporal patterns and attributes of highly numerous and disparate topics or themes within various genres. Moreover, the generalization capacities associated with the proposed model go beyond the context of Chinese literature and extend to cross-linguistic and cross-cultural analyses of sentiment and topics in historical documents. Because of such strong positive characteristics as performance, scalability, and flexibility for different domains and text types, the proposed model can become an effective tool for sentiment analysis in collecting ancient texts. It applies even in contemporary usage analysis, whereby context-induced ensemble intelligibility is pertinent in tasks such as literature or social media analysis. Furthermore, the outcomes of the corresponding iterations of the developed model, depicted in Fig 5, evidence its flexibility and effectiveness for various processes.

### 4.6  Discussion

This study proposed an unsupervised sentiment analysis model, combining BERT-based embeddings with GATs, significantly improving Chinese classical literature analysis. Traditionally, classical Chinese texts contain complex

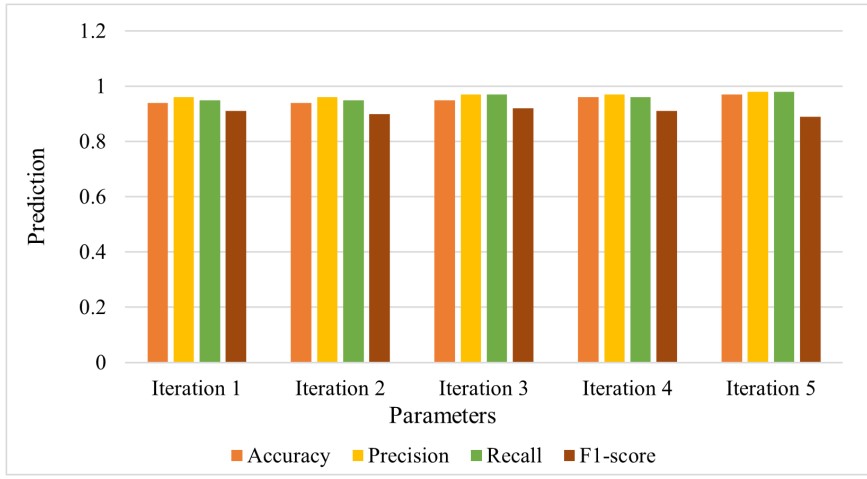

**Fig 5. Proposed model results on different iterations.**

long-range dependencies and contextual subtleties, and the model outperformed the aforementioned traditional methods, such as LSTMs and recurrent neural networks (RNNs). Its distinct feature is that it can work without labelled datasets, which is particularly appropriate in classical literature, where one often lacks data. The model effectively classifies sentiment via a graph-based approach with unsupervised learning, not requiring costly annotations. It leverages BERT embeddings marginally enhanced with sentiment data from specialized lexicons to interpret classical Chinese texts' complicated emotional and cultural qualities. Graph-based modelling through GATs was particularly effective in understanding word relationship-based texts. With GAT's attention mechanism, which selects what is most relevant from the word relationship, this method addresses classical Chinese's complex syntactic and semantic structure. Sentiment analysis was performed using unsupervised clustering, with K-Means-specific algorithms that helped group words with similar sentiment to increase their accuracy and depth. The computational efficiency of the model was a consideration of design. Unlike LSTM and RNN, which tend to be computationally expensive, especially with large datasets, our approach is scalable to large volumes of text, making it a more palatable solution for classical literature datasets of larger scales. The framework's scalability allows it to be used for Chinese classical texts (e.g., Tang dynasty dramas) and other sizeable unannotated text corpora across various domains. The model proposed for sentiment analysis of Chinese classical literature is promising. It addresses some key challenges, shows that such accuracy is scalable and efficient, and offers important insights about how sentiment can be classified in the case of such long texts. Yet, it also points out what can be refined and how it might be further developed. Fig 6 shows the result of the proposed VS Comparison models.

The performance compares of the proposed model with several state-of-the-art (SOTA) methods in sentiment analysis. Wang et al. [43] achieved strong performance with an accuracy of 0.94, precision of 0.93, recall of 0.95, and F1-score of 0.93. Zhong et al. [44] showed lower results, with accuracy of 0.84, precision of 0.82, recall of 0.87, and F1-score of 0.78, indicating a less effective model. Ihnaini et al. [40] reported accuracy of 0.91, precision of 0.89, recall of 0.92, and F1-score of 0.84, outperforming Zhong et al. but still lagging behind Wang et al. The proposed model, however, outperforms all comparison models with the highest accuracy of 0.95, precision of 0.97, recall 0.96, and F1-score 0.91, demonstrating its superior ability to classify sentiment in complex texts. Table 6 highlights the effectiveness of the proposed approach in sentiment analysis.

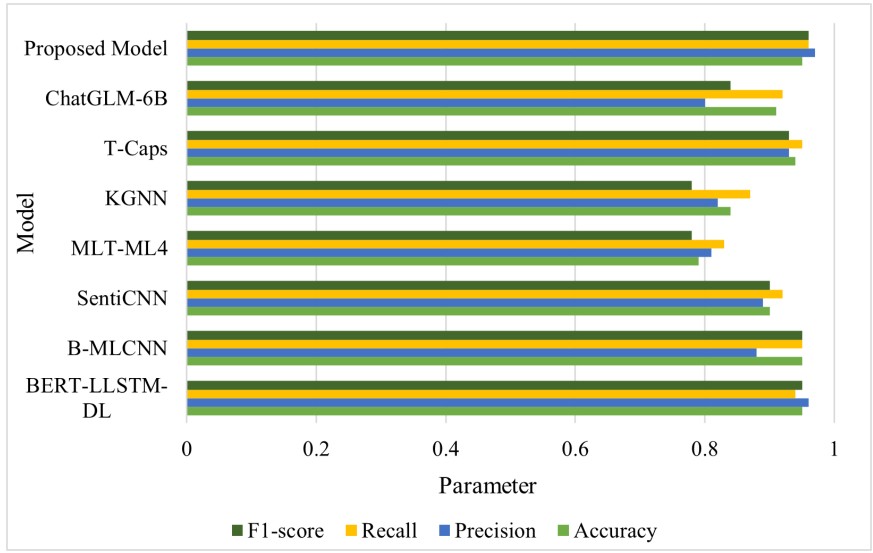

**Fig 6. Proposed VS comparison models result.**

**Table 6. The proposed model compares with state-of-the-art (SOTA) methods.**

| Study | Accuracy | Precision | Recall | F1-score |
|---|---|---|---|---|
| Wang et al [43] | 0.94 | 0.93 | 0.95 | 0.93 |
| Zhong et al [44] | 0.84 | 0.82 | 0.87 | 0.78 |
| Ihnaini et al [38] | 0.91 | 0.89 | 0.92 | 0.84 |
| **Proposed Model** | 0.95 | 0.97 | 0.96 | 0.91 |

## 5 Conclusion and future work

This study presents an innovative unsupervised sentiment analysis framework customized for Chinese classical literature based on the state-of-the-art deep learning model, such as BERT-based embeddings, Graph Attention Networks (GATs) and K-means clustering. The main difficulties of analyzing classical Chinese texts are that there is no annotated dataset and the complicated implicit expressions of emotion in the language. The traditional methods of sentiment analysis that involve supervised learning and labelled data struggle to capture the nuance and metaphor within these works of literature. To overcome these challenges, our framework exploits the use of BERT to generate contextualized word embeddings that can capture the meaning of a word in its specific context in the text. This step ensures that the model successfully interprets the complexities of classical Chinese, since the language context of the words significantly explains the meaning of words. Additionally, the model integrates the sentiment-explicit knowledge from the NTUSD sentiment lexicon to enhance the BERT embeddings to better understand the emotional content of the text. Graph Attention Networks (GATs) also bring an extra tinge of intricacy to the form of designing the relationship between the words in the text as nodes in a graph with co-occurrence and semantic dependencies as the edges. The attention mechanism present in GATs enables the model to pay more attention to the most useful word relationships, which boosts the quality of node representations and accuracy in sentiment classification. Scores of this approach over traditional sentiment analysis models, e.g., LSTM, BiLSTM and CNN, based on accuracy, precision, recall and F1-score are shown in the table below. The model successfully classifies the sentiments in Chinese classical literature into positive, negative, or neutral categories, revealing

the texts' emotional depth and cultural nuances. This work is a major improvement in using deep learning for literary analysis, providing a strong sentiment analysis tool for historical and culturally enriched domains. Although the results are very promising, there are several areas that one could improve and develop upon. A major limitation is that a fixed sentiment lexicon, such as NTUSD, is used, and all the emotional nuances and metaphors used in classical Chinese literature will be covered by this lexicon.

Although the presented model has demonstrated positive results, there are areas for improvement and future study. The use of a fixed sentiment Lexicon (e.g., NTUSD) that may not be able to exhaust the complicated feelings and metaphors implied in classical writings is also a limitation. Future research may include enriching and adjusting the lexicon to accommodate other emotions and cultural aspects. In addition, applying supervised learning techniques based on annotated datasets in a particular domain can improve the model's accuracy and robustness. Further studies could also test the use of the framework in other forms of historical literature and multilingual datasets, allowing for cross-linguistic sentiment analysis. Another line of development is the inclusion of fine-grained sentiment analysis that would enable sensing subtler emotional changes in longer texts.

## Supporting information

**S1 Dataset. Classical Chinese Texts Corpus: This corpus/repository includes the classical Chinese literature used in the experiments of sentiment analysis presented in this paper.** It contains the minimal set of data needed to reproduce the findings that have been presented in the manuscript/article, translated into English.
(ZIP)

**S2 Module-Wise Code Files. This package of Python scripts implements the proposed unsupervised sentiment analysis approach.** It contains a pre-processing script (preprocess.py), a script that calculates BERT embeddings (bert_encoder.py), a script to enrich embeddings with sentiment (enrich_embeddings.py), a script to build the graph (build_graph.py), a script to compute a graph attention module (gat_module.py), a script to do clustering and evaluation (clustering_and_evaluation.py), and utility/evaluation scripts. These files permit a complete replication of the experiments.
(ZIP)

**S3 LLM Comparison Code. There are Python notebooks and scripts to build a zero-shot sentiment classifier using ChatGPT-4o and DeepSeek r1.** These codes replicate the comparative evaluation experiments documented in the Results section.
(ZIP)

**S4 Model Output-JSON Files. All intermediate and final results produced by the proposed framework have been stored in this folder and include: Tokenized input samples, Enriched embeddings, Graph construction outputs, Cluster labels (positive, negative, neutral), Final cluster centroids, Performance metrics (accuracy, precision, recall, F1-score).** The results of these output directly contribute to the outcomes shown in Tables 4 and 5, and Figures 5 and 6.
(ZIP)

**S5 True Labels Reference File. A JSON file with ground truth sentiment labels used in evaluation and comparison to LLM prediction.**
(ZIP)

**S6 LLM Prediction Outputs. JSON files with the zero-shot sentiment classification predictions/outputs by ChatGPT-4o and DeepSeek r1.** These outputs were directly compared with our model.
(ZIP)

## Author contributions

**Data curation:** Xiaohan Yu, Jin Wang.

**Methodology:** Xiaohan Yu.

**Writing – original draft:** Xiaohan Yu, Jin Wang.

**Writing – review & editing:** Xiaohan Yu, Jin Wang.

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
