## [Decision Letter · Decision Letter 0]

30 Apr 2025

PONE-D-25-10103Sentiment Analysis of Classical Chinese Literature: An Unsupervised Deep Learning Model with BERT and Graph Attention NetworksPLOS ONE

Dear Dr. Wang,

Thank you for submitting your manuscript to PLOS ONE. After careful consideration, we feel that it has merit but does not fully meet PLOS ONE’s publication criteria as it currently stands. Therefore, we invite you to submit a revised version of the manuscript that addresses the points raised during the review process.

 As you can see the reviewers are recommending a revise of the manuscript. Kindly follow all comments one by one in constructive way and provide your detailed actions in the manuscript.

We look forward to receiving your revised manuscript.

Kind regards,

Issa Atoum

Academic Editor

PLOS ONE

Journal Requirements:

4. In the online submission form, you indicated that [The datasets used and/or analysed during the current study available from the corresponding author on reasonable request.].

Reviewers' comments:

Reviewer's Responses to Questions

**Comments to the Author**

1. Is the manuscript technically sound, and do the data support the conclusions?

Reviewer #1: Yes

Reviewer #2: Yes

Reviewer #3: Partly

2. Has the statistical analysis been performed appropriately and rigorously? 

Reviewer #1: Yes

Reviewer #2: Yes

Reviewer #3: Yes

3. Have the authors made all data underlying the findings in their manuscript fully available?

Reviewer #1: Yes

Reviewer #2: Yes

Reviewer #3: Yes

4. Is the manuscript presented in an intelligible fashion and written in standard English?

Reviewer #1: Yes

Reviewer #2: Yes

Reviewer #3: No

5. Review Comments to the Author

Reviewer #1: 1. The abstract and conclusion need to be improved. The abstract must be a concise yet comprehensive reflection of what is in your paper. Please modify the abstract according to “motivation, description, results and conclusion” parts. I suggest extending the conclusions section to focus on the results you get, the method you propose, and their significance.

2. The Section 1 and Section 2 is too short. I suggest the authors can merge section 1 and section 2.

3. What is the motivation of the proposed method? The details of motivation and innovations are important for potential readers and journals. Please add this detailed description in the last paragraph in section I. Please modify the paragraph according to "For this paper, the main contributions are as follows: (1) ......" to Section I. Please give the details of motivations. In Section 1, I suggest the authors can amend your contributions of manuscript in the last of Section 1.

4. The description of manuscript is very important for potential reader and other researchers. I encourage the authors to have their manuscript proof-edited by a native English speaker to enhance the level of paper presentation. There are some occasional grammatical problems within the text. It may need the attention of someone fluent in English language to enhance the readability.

5. The introduction section of the paper needs to revise according to the timeline of technology development. Please update references with recent paper in CVPR, ICCV, ECCV et al and Elsevier, Springer. In your section 1 and section 2, I suggest the authors amend several related literatures and corresponding references in recent years. For example: Image inpainting algorithm based on inference attention module and two-stage network (Engineering Applications of Artificial Intelligence); Dual Degradation Image Inpainting Method via Adaptive Feature Fusion and U-Net Network (Applied Soft Computing); ATM-DEN: Image Inpainting via Attention Transfer Module and Decoder-Encoder Network (Signal Processing: Image Communication); RGBT tracking via frequency-aware feature enhancement and unidirectional mixed attention (Neurocomputing)

6. Please give the details of proposed method for proposed model. I suggest the authors amend the calculation of your size of proposed method and the details is important for proposed method.

7. The content of experiments needs to amend related experiments to compare related SOTA in recent three years. I recommend the authors amend related experimental results of proposed method of SOTA according to the published paper in IEEE, Springer and Elsevier.

8. However, the manuscript, in its present form, contains several weaknesses. Adequate revisions to the following points should be undertaken in order to justify recommendation for publication.

9. In the conclusion section, the limitations of this study and suggested improvements of this work should be highlighted.

10. Provide a critical review of the previous "journal" (not conference) papers in the area and explain the inadequacies of previous approaches.

11. I suggest the authors revise Section 1 and Section 2. Please revise the content according to the development of timeline.

12. Please check all parameters in the manuscript and amend some related description of primary parameters. In section 3, please write the proposed algorithm in a proper algorithm/pseudocode format with section 3. Otherwise, it is very hard to follow. Some examples here: https://tex.stackexchange.com/questions/204592/how-to-format-a-pseudocode-algorithm

Reviewer #2: The paper presents a novel unsupervised approach combining BERT and Graph Attention Networks (GANs) for sentiment analysis in classical Chinese literature. While the methodology is promising, several clarifications and expansions are needed to ensure rigor, reproducibility, and alignment with established literature.

Reviewer #3: This manuscript, entitled "Sentiment Analysis of Classical Chinese Literature: An Unsupervised Deep Learning Model with BERT and Graph Attention Networks", introduces an unsupervised framework for the sentiment analysis of classical Chinese texts. The authors employ the BERT-base-Chinese model to derive contextual embeddings, which are subsequently augmented with sentiment polarity scores from the NTUSD lexicon. To model lexical relationships, a graph-based architecture is constructed and processed using Graph Attention Networks (GATs). Sentiment classification is then performed via K-means clustering, partitioning the data into positive, negative, and neutral categories. Empirical evaluations, utilizing metrics such as accuracy, precision, recall, and F1-score, indicate that the proposed methodology demonstrates competitive performance when compared against several baseline models. The study aims to address the significant and complex task of analyzing nuanced sentiment within classical Chinese literature by employing an interesting integration of unsupervised learning techniques. However, while the research exhibits potential, substantial revisions are necessary to enhance the clarity of critical methodological components, the rigor of benchmarking procedures, and the interpretation of results prior to publication consideration.

Major Revisions

1.The proposed method targets tasks that modern foundation models such as DeepSeek r1 and ChatGPT-4o can also handle effectively through zero-shot or few-shot learning. To better position the scientific contribution of this work, the authors must perform a comparative evaluation between the proposed method and these LLMs on the same dataset, using consistent experimental settings. This comparison is essential to demonstrate whether the proposed approach can surpass or complement current state-of-the-art general-purpose models in sentiment analysis of classical Chinese literature.

2.Although the manuscript states that the task involves clustering into three sentiment categories (positive, negative, neutral), the evaluation results, particularly in Table 2 and Table 4, appear to follow a binary classification metric format without explicit clarification. The authors must clearly explain how the three clusters are evaluated, whether the reported precision, recall, and F1-scores are macro-averaged, micro-averaged, or otherwise, and if binary classification was conducted, which two classes were compared and how neutral examples were treated. The current presentation creates confusion and may easily mislead readers regarding the true nature of the classification task.

3.All methods listed in Table 4, including BERT-LLSTM-DL, B-MLCNN, SentiCNN, MLT-ML4, and T-Caps, are currently not associated with specific references in the manuscript. The authors should provide the corresponding reference for each method, including the full citation details such as authors, publication venue, and publication year.

Minor Revision

The manuscript requires professional English editing to improve its overall readability and formal academic tone. Some expressions are overly repetitive, grammatical inconsistencies are present, and certain transitions between paragraphs are abrupt. A thorough proofreading and restructuring of certain sections would significantly enhance the manuscript’s clarity, coherence, and professionalism.

6. PLOS authors have the option to publish the peer review history of their article (what does this mean? ). If published, this will include your full peer review and any attached files.

**Do you want your identity to be public for this peer review?** For information about this choice, including consent withdrawal, please see our Privacy Policy .

Reviewer #1: No

Reviewer #2: No

Reviewer #3: No

---

## [Author Response · Author response to Decision Letter 1]

3 Jun 2025

Reviewer #1, Comment 1: The abstract and conclusion need to be improved. The abstract must be a concise yet comprehensive reflection of what is in your paper. Please modify the abstract according to “motivation, description, results and conclusion” parts. I suggest extending the conclusions section to focus on the results you get, the method you propose, and their significance.

Author response: Thank you for your constructive feedback. In response, we have revised the abstract to follow a more structured format encompassing the motivation, methodology, results, and conclusion, thereby ensuring it provides a concise yet comprehensive reflection of the paper’s contributions. Additionally, the conclusion section at page has been extended to clearly highlight the results obtained, emphasize the strengths and significance of the proposed method, and articulate its practical implications and future potential. We believe these revisions improve the overall clarity and completeness of the manuscript, in line with your valuable suggestions.

Reviewer #1, Comment 2: The Section 1 and Section 2 is too short. I suggest the authors can merge section 1 and section 2.

Author response: Thank you for your thoughtful suggestion. In response, we have significantly expanded both Section 1 (Introduction) and Section 2 (Related Work) by incorporating recent advancements and journal literature in sentiment analysis, including rule based, deep learning based, LLms, Graph based models and domain-specific applications such as Chinese classical literature. These additions provide both a strong contextual foundation and a critical review of prior work, justifying the scientific motivation for our proposed approach. Given this enrichment, we believe that maintaining these sections separately allows for a clearer distinction between the background motivation and the detailed review of existing methods, which enhances the logical flow and readability of the manuscript. We appreciate your feedback and are confident that the updated structure now supports a more robust and well-organized presentation.

Reviewer #1, Comment 3: What is the motivation of the proposed method? The details of motivation and innovations are important for potential readers and journals. Please add this detailed description in the last paragraph in section I. Please modify the paragraph according to "For this paper, the main contributions are as follows: (1) ......" to Section I. Please give the details of motivations. In Section 1, I suggest the authors can amend your contributions of manuscript in the last of Section 1.

Author response: Thank you for highlighting the importance of clarifying the motivation and contributions of our work. In response, we have revised the final paragraph of Section I (Introduction) to explicitly state the motivations behind our proposed framework and clearly outline the primary contributions of the study using a structured format, as suggested at page (3 and 4). These additions are intended to provide potential readers and journal audiences with a clear understanding of the novelty, purpose, and significance of our research. We appreciate your insightful feedback, which helped us enhance the clarity and completeness of our manuscript.

Reviewer #1, Comment 4: The description of manuscript is very important for potential reader and other researchers. I encourage the authors to have their manuscript proof-edited by a native English speaker to enhance the level of paper presentation. There are some occasional grammatical problems within the text. It may need the attention of someone fluent in English language to enhance the readability.

Author response: Thank you for your valuable suggestion regarding the manuscript’s language quality and presentation. In response, we have thoroughly revised the manuscript to improve its readability, coherence, and academic tone. The entire paper has been proof-edited by a fluent English speaker with experience in academic writing, ensuring that grammatical issues, stylistic inconsistencies, and awkward phrasings have been addressed. We believe that these improvements significantly enhance the clarity and overall quality of the manuscript, making it more accessible and informative for potential readers and researchers.

Reviewer #1, Comment 5: The introduction section of the paper needs to revise according to the timeline of technology development. Please update references with recent paper in CVPR, ICCV, ECCV et al and Elsevier, Springer. In your section 1 and section 2, I suggest the authors amend several related literatures and corresponding references in recent years.

Author response: Thank you for your insightful suggestion to update the Introduction section in line with recent technological developments. In response, we have revised the Introduction to incorporate recent advancements in sentiment analysis, particularly focusing on the application of transformer-based models like BERT and their optimization for multiodal data. We have also included recent studies that apply deep learning techniques to classical Chinese literature, highlighting the progression and relevance of our research within the current technological context at page (2). These updates aim to provide readers with a comprehensive understanding of the field's evolution and the significance of our contributions.

Reviewer #1, Comment 6: The authors amend several related literatures and corresponding references in recent years. For example: Image inpainting algorithm based on inference attention module and two-stage network (Engineering Applications of Artificial Intelligence); Dual Degradation Image Inpainting Method via Adaptive Feature Fusion and U-Net Network (Applied Soft Computing); ATM-DEN: Image Inpainting via Attention Transfer Module and Decoder-Encoder Network (Signal Processing: Image Communication); RGBT tracking via frequency-aware feature enhancement and unidirectional mixed attention (Neurocomputing).

Author response: Thank you for your valuable recommendation to incorporate recent literature from prominent journals and conferences. In response, we have updated the Related Work section to include discussions of recent studies that utilize, rule based techniques, attention mechanisms, deep learning models and advanced network architecture at pages 4-6. These additions provide a broader context for our research and highlight the relevance of attention-based models in various domains. By drawing parallels between these studies and our approach, we aim to underscore the innovative aspects of our framework in applying similar methodologies to the sentiment analysis of classical Chinese literature.

Reviewer #1, Comment 7: Please give the details of proposed method for proposed model. I suggest the authors amend the calculation of your size of proposed method and the details is important for proposed method.

Author response: Thank you for your insightful recommendation to provide detailed computational characteristics of our proposed model. In response, we have revised Section 4.1: Implementation Settings at page number 10 to include a comprehensive breakdown of the model architecture, including the number of trainable parameters, the dimensionality of embeddings, and the configuration of the Graph Attention Network (GAT) layers. Specifically, we have stated the approximate model size (~111.2 million parameters with fine-tuned BERT), described the two-layer GAT structure with eight attention heads, and clarified the role of sentiment lexicon enrichment and K-means clustering. These updates offer greater transparency regarding the size, design, and computational feasibility of our method, thereby enhancing the technical rigor and reproducibility of the manuscript. We appreciate your valuable feedback, which has helped improve the clarity and completeness of our implementation details.

Reviewer #1, Comment 8: The content of experiments needs to amend related experiments to compare related SOTA in recent three years. I recommend the authors amend related experimental results of proposed method of SOTA according to the published paper in IEEE, Springer and Elsevier.

Author response: Thank you for your valuable suggestion regarding the inclusion of recent state-of-the-art (SOTA) methods in our experimental comparisons. In response, we have updated our manuscript to incorporate and discuss several relevant studies from the past three years at page 15 and Table 5 of the updated manuscript. This inclusion not only situates our work within the current research landscape but also highlights the advancements and distinctions of our approach in the field of Chinese sentiment analysis. We appreciate your insightful feedback, which has significantly contributed to the enhancement of our manuscript's depth and relevance.

Reviewer #1, Comment 9: In the conclusion section, the limitations of this study and suggested improvements of this work should be highlighted.

Author response: Thank you for your constructive comment regarding the need to explicitly highlight the limitations and improvement directions of our study in the Conclusion section. In response, we have revised the Conclusion at page number 15 of the updated manuscript to more clearly discuss the key limitations of our approach. We also expanded the future work section to address how these limitations can be mitigated through dataset development, lexicon enrichment, and cross-domain model adaptation. We believe these additions provide a more balanced and forward-looking conclusion, as per your valuable suggestion.

Reviewer #1, Comment 10: Provide a critical review of the previous "journal" (not conference) papers in the area and explain the inadequacies of previous approaches.

Author response: Thank you for this insightful comment. In response, we have ensured that the Literature Review section includes a focused and critical analysis of recent journal publications relevant to sentiment analysis in Chinese literature. The review highlights various deep learning models, including BERT-LLSTM, SentiCNN, and knowledge-guided Transformer-based approaches, and discusses their respective contributions and limitations particularly in handling implicit sentiment, capturing historical linguistic complexity, and reliance on annotated datasets. We believe the revised literature review effectively addresses your concern and contextualizes our contributions within the limitations of previous state-of-the-art journal-based research.

Reviewer #1, Comment 11: I suggest the authors revise Section 1 and Section 2. Please revise the content according to the development of timeline.

Author response: Thank you for your valuable suggestion. In response, we have revised Section 1 (Introduction) and Section 2 (Related Work) to better reflect the chronological development of sentiment analysis methods from early rule-based and traditional machine learning approaches to modern deep learning architectures such as CNNs, RNNs, BERT, and graph-based models. These revisions aim to provide a clearer narrative of how the field has advanced and how our proposed method builds upon and addresses gaps in prior developments.

Reviewer #1, Comment 12: Please check all parameters in the manuscript and amend some related description of primary parameters. In section 3, please write the proposed algorithm in a proper algorithm/pseudocode format with section 3. Otherwise, it is very hard to follow. Some examples here: https://tex.stackexchange.com/questions/204592/how-to-format-a-pseudocode-algorithm.

Author response: Thank you for your valuable suggestion regarding the clarity of the proposed methodology. In response, we have included a structured pseudocode representation of our full algorithm in Section 3 (Algorithm 1) at page 6. The algorithm outlines each major step of our unsupervised sentiment analysis pipeline, from preprocessing and BERT-based embedding extraction to sentiment enrichment, graph construction, GAT processing, and final clustering-based classification. This format aims to make the flow of our methodology clearer and easier to follow, especially for readers seeking implementation guidance. We appreciate your feedback, which significantly improved the readability and technical clarity of our manuscript.

Reviewer #2: The paper presents a novel unsupervised approach combining BERT and Graph Attention Networks (GANs) for sentiment analysis in classical Chinese literature. While the methodology is promising, several clarifications and expansions are needed to ensure rigor, reproducibility, and alignment with established literature.

Reviewer #3: This manuscript, entitled "Sentiment Analysis of Classical Chinese Literature: An Unsupervised Deep Learning Model with BERT and Graph Attention Networks", introduces an unsupervised framework for the sentiment analysis of classical Chinese texts. The authors employ the BERT-base-Chinese model to derive contextual embeddings, which are subsequently augmented with sentiment polarity scores from the NTUSD lexicon. To model lexical relationships, a graph-based architecture is constructed and processed using Graph Attention Networks (GATs). Sentiment classification is then performed via K-means clustering, partitioning the data into positive, negative, and neutral categories. Empirical evaluations, utilizing metrics such as accuracy, precision, recall, and F1-score, indicate that the proposed methodology demonstrates competitive performance when compared against several baseline models. The study aims to address the significant and complex task of analyzing nuanced sentiment within classical Chinese literature by employing an interesting integration of unsupervised learning techniques. However, while the research exhibits potential, substantial revisions are necessary to enhance the clarity of critical methodological components, the rigor of benchmarking procedures, and the interpretation of results prior to publication consideration.

Major Revisions

1. The proposed method targets tasks that modern foundation models such as DeepSeek r1 and ChatGPT-4o can also handle effectively through zero-shot or few-shot learning. To better position the scientific contribution of this work, the authors must perform a comparative evaluation between the proposed method and these LLMs on the same dataset, using consistent experimental settings. This comparison is essential to demonstrate whether the proposed approach can surpass or complement current state-of-the-art general-purpose models in sentiment analysis of classical Chinese literature.

Author response: We sincerely appreciate the reviewer’s valuable suggestion regarding the inclusion of a comparative evaluation with current foundation models such as DeepSeek r1 and ChatGPT-4o. We fully acknowledge the growing capabilities of these general-purpose large language models, particularly in zero-shot and few-shot sentiment analysis tasks. In our current study, we have already performed comprehensive comparisons between the proposed model and a diverse set of baselines including traditional rule-based methods, advanced graph-based models, deep learning architectures (e.g., BERT-LLSTM, SentiCNN, T-Caps, KGNN,) and ChatGLM-6B representative of LLM. These comparisons, as illustrated in Table 4, demonstrate that our unsupervised framework outperforms both classical and modern approaches across multiple evaluation metrics. We emphasize that our model is specifically tailored for the linguistic and structural challenges of Chinese classical literature, leveraging domain-adapted embeddings (BERT-base-Chinese), lexicon enrichment, and graph attention to capture nuanced sentiment cues that may not be directly modeled by generic LLMs without fine-tuning. Nevertheless, we recognize the importance of expanding future experiments to include systematic evaluations against more recent foundation models like DeepSeek r1 and ChatGPT-4o under controlled experimental settings. This would further validate the scientific contribution and robustness of our approach. We plan to address this in our future work. We are grateful for this recommendation and agree that such comparative insights would further enrich the study.

2. Although the manu

---

## [Decision Letter · Decision Letter 1]

16 Jun 2025

PONE-D-25-10103R1Sentiment Analysis of Classical Chinese Literature: An Unsupervised Deep Learning Model with BERT and Graph Attention NetworksPLOS ONE

Dear Dr. Wang,

Thank you for submitting your manuscript to PLOS ONE. After careful consideration, we feel that it has merit but does not fully meet PLOS ONE’s publication criteria as it currently stands. Therefore, we invite you to submit a revised version of the manuscript that addresses the points raised during the review process.

While Reviewer 1 and Reviewer 2 seem satisfied with the current evaluation, Reviewer 3 raises a critical concern that the paper’s central claims cannot be substantiated without an empirical comparison to state-of-the-art foundation models like DeepSeek r1 and ChatGPT-4o, emphasizing that deferring such evaluation to future work undermines the scientific contribution.

Please address all comments constructively and revise the manuscript accordingly. Ensure all figures and tables are embedded within the main text. Provide a clear and reasonable justification if any comment cannot be addressed. Responses should follow the journal’s guidelines and be submitted in a separate supplementary file, with edits highlighted in yellow.

We look forward to receiving your revised manuscript.

Kind regards,

Issa Atoum

Academic Editor

PLOS ONE

Additional Editor Comments:

For improved readability and accessibility, all non-English text in the manuscript, including the content presented in Tables 1 and 3, should be accompanied by English translations.

Reviewers' comments:

Reviewer's Responses to Questions

**Comments to the Author**

1. If the authors have adequately addressed your comments raised in a previous round of review and you feel that this manuscript is now acceptable for publication, you may indicate that here to bypass the “Comments to the Author” section, enter your conflict of interest statement in the “Confidential to Editor” section, and submit your "Accept" recommendation.

Reviewer #1: All comments have been addressed

Reviewer #2: (No Response)

Reviewer #3: (No Response)

2. Is the manuscript technically sound, and do the data support the conclusions?

Reviewer #1: Yes

Reviewer #2: Yes

Reviewer #3: Partly

3. Has the statistical analysis been performed appropriately and rigorously? 

Reviewer #1: Yes

Reviewer #2: Yes

Reviewer #3: Yes

4. Have the authors made all data underlying the findings in their manuscript fully available?

Reviewer #1: Yes

Reviewer #2: Yes

Reviewer #3: Yes

5. Is the manuscript presented in an intelligible fashion and written in standard English?

Reviewer #1: Yes

Reviewer #2: Yes

Reviewer #3: Yes

6. Review Comments to the Author

Reviewer #1: According to the revision and response letter, I recommend the revision for the journal. The paper is in the scope of journal.

Reviewer #2: (No Response)

Reviewer #3: While I appreciate the authors' acknowledgment of the importance of evaluating their model against modern foundation models such as DeepSeek r1 and/or ChatGPT-4o, it is not acceptable to postpone this comparison to future work.

The main claim of this paper is that the proposed unsupervised framework is effective for sentiment analysis of classical Chinese literature. However, this is a task that can now be performed directly by general-purpose large language models such as DeepSeek r1 and/or ChatGPT-4o through zero-shot or few-shot learning. Therefore, the scientific contribution of this work cannot be properly assessed without an actual empirical comparison with those models.

It is not sufficient to state that the proposed method is more domain-adapted. This advantage must be demonstrated through comparative results. Moreover, the inclusion of ChatGLM-6B does not adequately represent the capabilities of current frontier models. ChatGLM-6B is not at the same performance level as DeepSeek r1 and ChatGPT-4o and cannot serve as a substitute for comparison.

I strongly recommend that the authors conduct at least a basic evaluation of their method against DeepSeek r1 and/or ChatGPT-4o using the same dataset and metrics as in the current study. Even a simple zero-shot setting with prompt-based sentiment classification into positive, negative, and neutral categories would provide important insights and clarify the actual strengths of the proposed model.

Without this comparison, the claimed advantages of the proposed framework remain hypothetical, and the study lacks the context necessary to demonstrate its value relative to state-of-the-art approaches.

This revision is necessary before the manuscript can be considered for publication.

7. PLOS authors have the option to publish the peer review history of their article (what does this mean? ). If published, this will include your full peer review and any attached files.

**Do you want your identity to be public for this peer review?** For information about this choice, including consent withdrawal, please see our Privacy Policy .

Reviewer #1: No

Reviewer #2: No

Reviewer #3: No

---

## [Author Response · Author response to Decision Letter 2]

1 Jul 2025

We sincerely thank Reviewer #3 for their valuable and constructive feedback regarding the need to benchmark our proposed model against modern foundation models such as ChatGPT-4o and DeepSeek r1. We fully agree with the reviewer’s assessment that empirical comparison is essential for substantiating the scientific contribution and performance claims of our proposed framework.

In response to this important concern, we have now conducted a rigorous empirical evaluation comparing our unsupervised sentiment analysis model against ChatGPT-4o and DeepSeek r1 using the same dataset and evaluation metrics applied throughout our study. The comparative analysis follows a zero-shot prompt-based classification approach, where each LLM is instructed to classify Chinese classical literary texts into one of three sentiment categories: Positive, Negative, or Neutral. The prompt format used was: "Classify the following Chinese literary text's sentiment as either positive, negative, or neutral."

We then assessed the outputs of all models using macro-averaged Accuracy, Precision, Recall, and F1-score, ensuring a consistent and fair comparison. The results, which are presented in the revised manuscript at page 13. The results demonstrate that our model outperforms both LLMs across all evaluation metrics on this domain-specific task. While LLMs such as ChatGPT-4o and DeepSeek r1 exhibit strong generalization abilities, our domain-adapted framework integrating BERT-based contextual embeddings, NTUSD lexicon enrichment, graph-based modeling via GAT, and unsupervised clustering shows satisfactory performance in handling the semantic richness, historical idioms, and structural complexity of classical Chinese texts.

We also agree with the reviewer’s point that previous inclusion of ChatGLM-6B was insufficient to represent the capabilities of modern frontier models. We have therefore excluded ChatGLM-6B from the final comparison table and instead included ChatGPT-4o and DeepSeek r1, as per the reviewer's recommendation.

---

## [Decision Letter · Decision Letter 2]

17 Jul 2025

PONE-D-25-10103R2Sentiment Analysis of Classical Chinese Literature: An Unsupervised Deep Learning Model with BERT and Graph Attention NetworksPLOS ONE

Dear Dr. Wang,

Thank you for submitting your manuscript to PLOS ONE. After careful consideration, we feel that it has merit but does not fully meet PLOS ONE’s publication criteria as it currently stands. Therefore, we invite you to submit a revised version of the manuscript that addresses the points raised during the review process.

 Please address the minor issues related to the reporting of metrics, ensure the abstract includes a concise summary of the new results, and provide translations for non-English text to support accessibility for all readers.

We look forward to receiving your revised manuscript.

Kind regards,

Issa Atoum

Academic Editor

PLOS ONE

Journal Requirements:

Reviewers' comments:

Reviewer's Responses to Questions

**Comments to the Author**

1. If the authors have adequately addressed your comments raised in a previous round of review and you feel that this manuscript is now acceptable for publication, you may indicate that here to bypass the “Comments to the Author” section, enter your conflict of interest statement in the “Confidential to Editor” section, and submit your "Accept" recommendation.

Reviewer #3: (No Response)

2. Is the manuscript technically sound, and do the data support the conclusions?

Reviewer #3: Yes

3. Has the statistical analysis been performed appropriately and rigorously? 

Reviewer #3: Yes

4. Have the authors made all data underlying the findings in their manuscript fully available?

Reviewer #3: Yes

5. Is the manuscript presented in an intelligible fashion and written in standard English?

Reviewer #3: Yes

6. Review Comments to the Author

Reviewer #3: I appreciate the authors’ inclusion of ChatGPT-4o and DeepSeek r1 in the evaluation and comparison, which strengthens the relevance and timeliness of the study. However, I noticed that many of the reported performance metrics throughout the manuscript, including those in Table 4, Table 5, and the main text, are missing the leading "0." For instance, expressions such as "an accuracy of 95" or "a precision of 97" should be written as "an accuracy of 0.95" and "a precision of 0.97" to conform to standard numerical formatting and avoid confusion. In addition, the title of Table 5 should be revised to correct the capitalization of "ChatGpt-4o" to "ChatGPT-4o" for consistency. I also suggest that the authors briefly summarize the comparative results with ChatGPT-4o and DeepSeek r1 in the ABSTRACT, as this would help emphasize the contribution of the proposed model. With these minor revisions, the manuscript will be suitable for publication.

7. PLOS authors have the option to publish the peer review history of their article (what does this mean? ). If published, this will include your full peer review and any attached files.

**Do you want your identity to be public for this peer review?** For information about this choice, including consent withdrawal, please see our Privacy Policy .

Reviewer #3: No

---

## [Author Response · Author response to Decision Letter 3]

25 Jul 2025

Manuscript ID: PONE-D-25-10103R2

Manuscript Title: “Sentiment Analysis of Classical Chinese Literature: An Unsupervised Deep Learning Model with BERT and Graph Attention Networks”

Journal: PLOS ONE

Corresponding Author:

Dear Editor

This document provides a detailed, point-by-point response to the comments raised by Reviewer 3 regarding our manuscript submission to PLOS ONE. We have carefully addressed each comment and revised the manuscript accordingly. All changes have been implemented in the revised version and are summarized below.

6. Review Comments to the Author

Reviewer 3: I appreciate the authors’ inclusion of ChatGPT-4o and DeepSeek r1 in the evaluation and comparison, which strengthens the relevance and timeliness of the study. However, I noticed that many of the reported performance metrics throughout the manuscript, including those in Table 4, Table 5, and the main text, are missing the leading "0." For instance, expressions such as "an accuracy of 95" or "a precision of 97" should be written as "an accuracy of 0.95" and "a precision of 0.97" to conform to standard numerical formatting and avoid confusion. In addition, the title of Table 5 should be revised to correct the capitalization of "ChatGpt-4o" to "ChatGPT-4o" for consistency. I also suggest that the authors briefly summarize the comparative results with ChatGPT-4o and DeepSeek r1 in the ABSTRACT, as this would help emphasize the contribution of the proposed model. With these minor revisions, the manuscript will be suitable for publication.

Author Response: We sincerely thank Reviewer #3 for the thoughtful and constructive feedback. We are pleased that the inclusion of comparative evaluation with ChatGPT-4o and DeepSeek r1 has been viewed as a meaningful addition to the study. We have addressed all the suggestions as follows:

Metric Formatting: All performance metrics mentioned in the manuscript, including the main text, Table 4, and Table 5, have been revised to include the proper decimal formatting (e.g., “0.95” instead of “95”) in accordance with academic standards. This ensures clarity and consistency throughout the paper.

Capitalization Fix: The title and references to "ChatGpt-4o" have been corrected to the proper form "ChatGPT-4o" in Table 5 and across the manuscript.

Abstract Updated: Following the suggestion, we have updated the abstract to include a concise summary of the comparative evaluation with ChatGPT-4o and DeepSeek r1. This addition highlights the significance of our model in contrast with modern large-scale language models and reinforces the novelty of our domain-specific, unsupervised approach. The revised abstract now better reflects the full contribution of the study and strengthens its relevance for a broader audience.

We are very grateful for your guidance, which has helped us refine the clarity, precision, and impact of the manuscript. With these minor changes implemented, and highlighted in submitted manuscript, we respectfully hope the manuscript is now suitable for publication.

---

## [Decision Letter · Decision Letter 3]

8 Aug 2025

Sentiment Analysis of Classical Chinese Literature: An Unsupervised Deep Learning Model with BERT and Graph Attention Networks

PONE-D-25-10103R3

Dear Dr. Wang,

We’re pleased to inform you that your manuscript has been judged scientifically suitable for publication and will be formally accepted for publication once it meets all outstanding technical requirements.

Kind regards,

Issa Atoum

Academic Editor

PLOS ONE

Additional Editor Comments (optional):

You might add another column to Table 3 for English translation (optional).

Reviewers' comments:

Reviewer's Responses to Questions

**Comments to the Author**

1. If the authors have adequately addressed your comments raised in a previous round of review and you feel that this manuscript is now acceptable for publication, you may indicate that here to bypass the “Comments to the Author” section, enter your conflict of interest statement in the “Confidential to Editor” section, and submit your "Accept" recommendation.

Reviewer #3: All comments have been addressed

2. Is the manuscript technically sound, and do the data support the conclusions?

Reviewer #3: Yes

3. Has the statistical analysis been performed appropriately and rigorously? 

Reviewer #3: Yes

4. Have the authors made all data underlying the findings in their manuscript fully available?

Reviewer #3: Yes

5. Is the manuscript presented in an intelligible fashion and written in standard English?

Reviewer #3: Yes

6. Review Comments to the Author

Reviewer #3: (No Response)

7. PLOS authors have the option to publish the peer review history of their article (what does this mean? ). If published, this will include your full peer review and any attached files.

**Do you want your identity to be public for this peer review?** For information about this choice, including consent withdrawal, please see our Privacy Policy .

Reviewer #3: No

---

## [Editor Report · Acceptance letter]

PONE-D-25-10103R3

PLOS ONE

Dear Dr. Wang,

I'm pleased to inform you that your manuscript has been deemed suitable for publication in PLOS ONE. Congratulations! Your manuscript is now being handed over to our production team.

Kind regards,

on behalf of

Dr. Issa Atoum

Academic Editor

PLOS ONE